# Machining water through laser cutting of nanoparticle-encased water pancakes

Jicheng Niu[1,2], Wenjing Liu[3], Jasmine Xinze Li[2], Xianglong Pang[3], Yulin Liu[1,2], Chao Zhang[1,2], Keyang Yue[1,2], Yulin Zhou[1,2], Feng Xu [1,2], Xiaoguang Li [3] ✉ & Fei Li [1,2] ✉

Due to the inherent disorder and fluidity of water, precise machining of water through laser cutting are challenging. Herein we report a strategy that realizes the laser cutting machining of water through constructing hydrophobic silica nanoparticle-encased water pancakes with sub-millimeter depth. Through theoretical analysis, numerical simulation, and experimental studies, the developed process of nanoparticle-encased water pancake laser cutting and the parameters that affect cutting accuracy are verified and elucidated. We demonstrate that laser-fabricated water patterns can form diverse self-supporting chips (SSCs) with openness, transparency, breathability, liquid morphology, and liquid flow control properties. Applications of laser-fabricated SSCs to various fields, including chemical synthesis, biochemical sensing, liquid metal manipulation, patterned hydrogel synthesis, and drug screening, are also conceptually demonstrated. This work provides a strategy for precisely machining water using laser cutting, addressing existing laser machining challenges and holding significance for widespread fields involving fluid patterning and flow control in biological, chemical, materials and biomedical research.

Water is an essential natural resource with applications in innumerable scientific disciplines. In recent years, research on control methods for the patterning and flow of small volumes of water has aroused extensive interest in the fields of material science[1], biology[2], chemistry[3] and medicine[4,5]. However, the machining of water through precise cutting remains a great challenge, due to water's inherent disorder and fluidity. Just as an ancient poem says: "Draw a knife, cut off water and flow more", that is, scratching water with a solid tool, the scratches will heal quickly.

Laser-processing based technologies (e.g., laser cutting, engraving, lithography, and printing), which hold advantages of high precision, fast speed manufacturing, and operational simplicity, have been widely employed in industrial production[6], electronics processing[7,8], and micromachining[9,10]. For example, fabrication of solid microfluidic channels has been achieved by laser cutting, printing and pyrolysis[9,10]. However, direct machining of water using lasers remains challenging, since laser radiation pressure can only create weak deformation and depression of the water surface and is too weak to overcome the combined effects of gravity and water surface tension[11,12]. Cutting water using the thermal effects of lasers, which is similar to laser cutting of solid materials, is an alternative strategy. However, this strategy is so far impossible to implement due to water's intrinsic properties of high light transmission, fluidity, and surface energy. First, the high light transmittance of water causes water to absorb laser energy with poor efficiency, limiting the amount of heat the laser provides to cut water through. Second, laser cutting of water involves more complicated physical behaviors than processing of solid due to its fluidity and high surface energy, leading to difficulties in generating complex and finely

[1]The Key Laboratory of Biomedical Information Engineering of Ministry of Education, School of Life Science and Technology, Xi'an Jiaotong University, Xi'an 710049, P.R. China. [2]Bioinspired Engineering and Biomechanics Center (BEBC), Xi'an Jiaotong University, Xi'an 710049, P.R. China. [3]Shaanxi Basic Discipline (Liquid Physics) Research Center, School of Physical Science and Technology, Northwestern Polytechnical University, Xi'an, China.
✉e-mail: lixiaoguang@nwpu.edu.cn; feili@mail.xjtu.edu.cn

patterned structures. Therefore, there is still an unmet need for an effective strategy for laser-based machining of water.

Herein, we developed a strategy for machining water through laser cutting of nanoparticle-encased water pancakes (LCNEWP), in which water was coated with hydrophobic $SiO_2$ nanoparticles (NPs) and manipulated to form a water pancake with sub-millimeter thickness. Through this strategy, water pancakes can be successfully cut to generate submillimeter-scale cutting slits. The process of LCNEWP and the parameters that affect the cutting accuracy were investigated through theoretical analysis, numerical simulation, and experimental study. Then, under experimental conditions that optimized water pancake thickness and laser power, various complicated water patterns (e.g., text, drawings, and complex channels) were produced, and a variety of liquid manipulations were implemented using SSCs formed by patterned water, providing favorable conditions for a variety of practical applications. The applications of SSCs in the fields of chemical synthesis, biochemical sensing, smart material synthesis, and biomedicine were also conceptually demonstrated. As such, our work provides a

strategy to overcome the challenge of laser cutting to precisely machine water and a path to achieve water patterning in open environments in an easy and rapid manner, holding a wide range of application potentials in chemistry, biology, materials science, and biomedicine.

## Results

### Machining of nanoparticle-encased water pancakes by laser cutting

To cut water with a laser, we minimized the effect of properties of water that make laser cutting challenging, including light transmittance, fluidity, and surface energy. As illustrated in Fig. 1a, water was first added to and rolled in a Petri dish coated with a sol-gel film consisting of hydrophobic $SiO_2$ NPs. As such film has weak binding forces, the rolled water became coated with a $SiO_2$ nanoparticulate shell[13], forming a non-wetting water pancake. Greater than 95% of the water was then withdrawn from the nanoparticle-encased water pancake (NEWP). During the water extraction process, the thickness and apparent surface area of the NEWP gradually decreased. The reduced

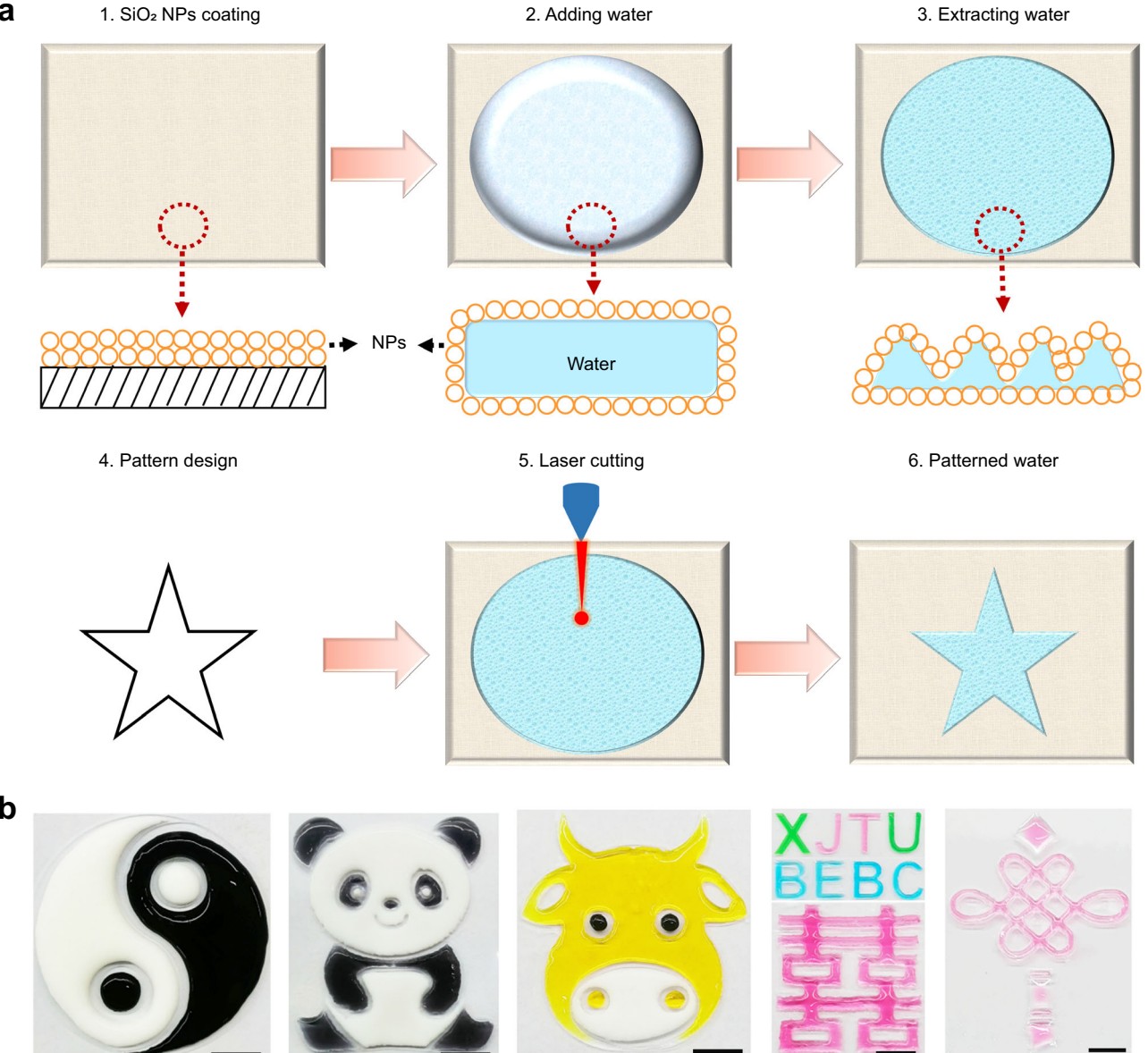

**Fig. 1 | Machining of nanoparticle-encased water pancakes through laser cutting. a** Schematic illustration of machining water through laser cutting of water pancake encased by silica nanoparticles (NPs). **b** Drawings, Chinese characters and letters formed by injecting various liquids (ink, milk, and pigment) into water patterned by laser cutting. The scale bars represent 1 cm.

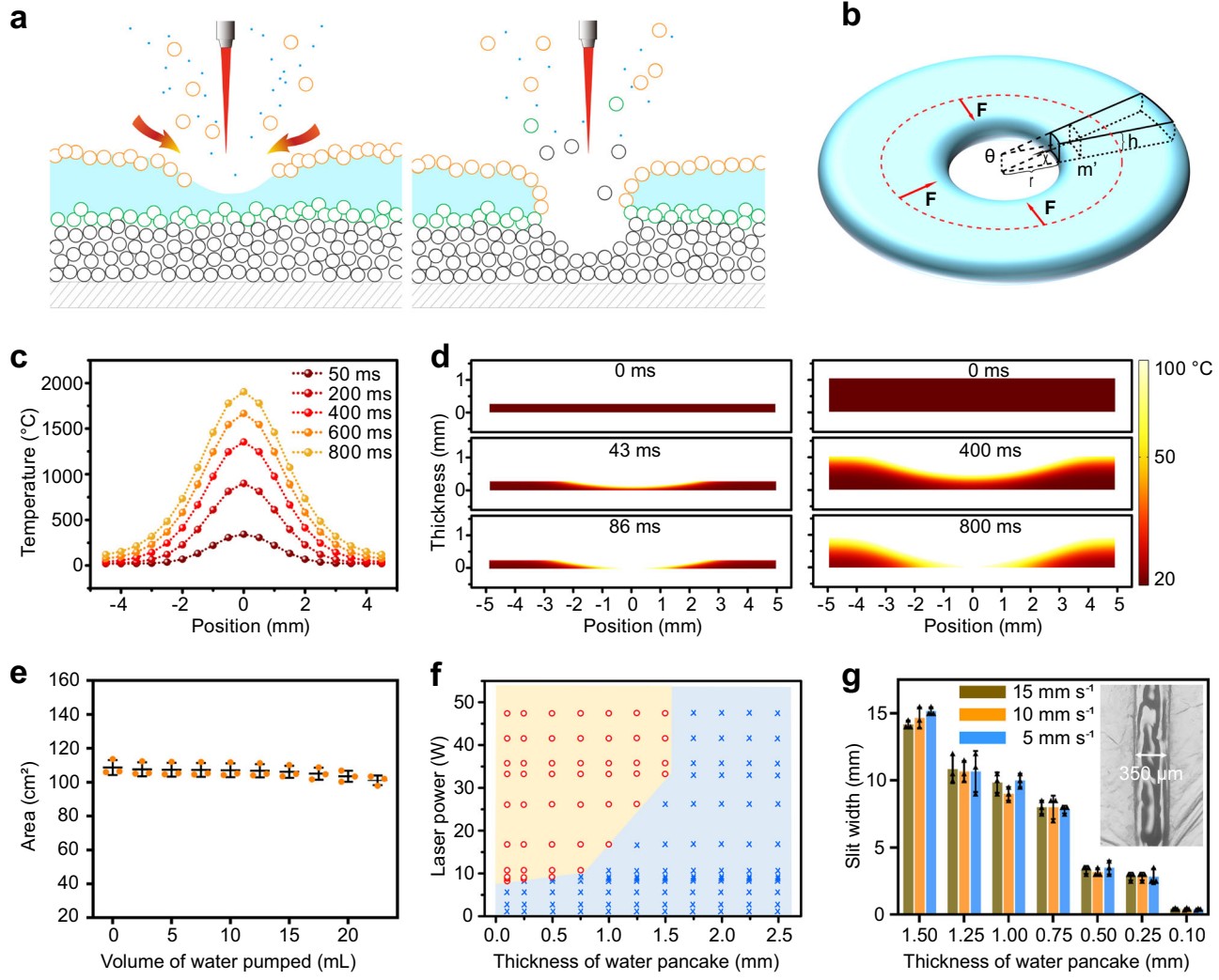

**Fig. 2 | Theoretical analysis, numerical simulation and experimental optimization of parameters for laser cutting of nanoparticle-encased water pancakes.** **a** Schematic diagram of LCNEWP. **b** Theoretical model of liquid flow during laser cutting. **c** Simulation of temperature of the SiO$_2$ nanoparticles around the laser spot after exposure to the laser. **d** Simulation of water shape evolution and temperature distribution during laser cutting, the initial thicknesses of the water in water pancake are 0.25 mm and 1.00 mm, respectively. **e** Variation of NEWP top view area with volume of water being pumped out (with an initial water volume of 25 mL). Data of distinct samples are presented as mean ± s.d. $n = 3$. **f** Effect of NEWP thickness and laser power on the outcome of LCNEWP. Brown symbols and background represent values at which the NEWP can be cut through by the laser. Blue symbols and background represent values at which the NEWP cannot be cut through by the laser. **g** Effect of NEWP thickness and cutting speed on slit width. Data of distinct samples are presented as mean ± s.d. $n = 3$. Insert image is the optical microscopic photograph of LCNEWP.

surface area of the NEWP causes interfacial NP jamming, resulting in a crumpled surface layer visible to the naked eye. Since the NP film not only enhances the absorption of laser energy but also prevents the water surface from shrinking under the action of surface energy at the incision after the water is cut, processing of water into the thin NEWP allows the achievement of high precision laser cutting. We then demonstrated this strategy in the fabrication of various patterns, including drawings, Chinese characters, and letters, through LCNEWP and the injection of colored liquids (ink, milk, and pigment) into NEWPs (Fig. 1b).

**Theoretical analysis, numerical simulation and optimized experimental parameters for laser cutting of nanoparticle-encased water pancakes**

We next developed a theoretical model to reveal the mechanism of LCNEWP (Fig. 2a), involving two main physical processes, water vaporization and water replenishment. First, after coming into contact with a laser with wavelength of 10.6 μm, the SiO$_2$ NPs on the surface of the water pancake absorb laser energy (Supplementary Fig. 1). Then,

the temperature of SiO$_2$ NPs increases and thermal energy is transferred to the underlying water, causing water around the laser irradiation spot to evaporate. Meanwhile, the surrounding water flows towards the laser irradiated location to replace the vaporized water, bringing NPs to the exposed water surface. When the speed of water vaporization is greater than the speed of water replenishment, the NEWP can be cut through. As the hydrophobic NPs coat the exposed water surface again, the incision cannot be healed through water flow, leading to a clean cut through the NEWP.

First, to elucidate the physical processes involved in LCNEWP, we performed theoretical analysis of both LCNEWP mechanics and its heat transfer process. For a given cutting procedure, let the thickness of water be $h$ and the radius of incision created by the cut be $r$ (Fig. 2b). The process of LCNEWP is simulated as the water occupying a volume of $\pi r^2 h$ is vaporized and the surrounding water flows into the vacant cylinder formed by this vaporization from the side of the cylinder. Three types of energy are involved in the process of water vaporization: the energy absorbed by the NPs and transferred to the water ($Q_q$), the energy consumed as the water heats ($Q_h$), and the energy

consumed during the liquid-vapor phase transition ($Q_v$). $Q_q$, $Q_h$, and $Q_v$ are defined by Eqs. (1–3).

$$Q_q = q \times s \times t_0 \tag{1}$$

$$Q_h = c \times m \times \Delta T = c\rho sh\Delta T \tag{2}$$

$$Q_v = \rho sh L_v \tag{3}$$

where $q$ is the heat flux transferred from the surface particles to the water, $s$ is the plane area of the vaporized part of the water, $t_0$ is the duration of water vaporization, $c$ is the specific heat capacity of the water, $m$ is the mass of the vaporized water, $\Delta T$ is the water's temperature change, $\rho$ is the water density, and $L_v$ is the latent heat of vaporization during the liquid-vapor phase transition. When $Q_q \geq Q_h + Q_v$, a cut is achieved. Under this condition, the vaporization time ($t_0$) is positively correlated with $h$, but negatively correlated with $q$, as shown in Eq. (4).

$$t_0 \geq h \frac{c\rho\Delta T + \rho L_V}{q} \tag{4}$$

During water replenishment in the LCNEWP process, the flow of surrounding water towards the incision is mainly driven by the internal pressure of water ($\mathbf{F}$) (Fig. 2b). The uniform acceleration of a rigid body with the same mass ($m$) as the incision area driven by pressure is used to simulate the flow process of the water during LCNEWP. $\mathbf{F}$, $m$, and $\mathbf{a}$ are represented by the following Eqs. (5–7).

$$\mathbf{F} = \mathbf{p}s' = \rho\mathbf{g}\frac{h}{2} \times 2\pi rh = \pi rh^2\rho\mathbf{g} \tag{5}$$

$$m = \pi r^2 h\rho \tag{6}$$

$$\mathbf{a} = \frac{\mathbf{F}}{m} = \frac{\mathbf{g}h}{r} \tag{7}$$

where $\mathbf{p}$ is the pressure, $s'$ is the surface area over which the pressure is applied, $\mathbf{g}$ is the acceleration of gravity, and $\mathbf{a}$ is the acceleration of the rigid body towards the incision. To replenish the incision area with the surrounding water, the center of gravity of the rigid body needs to move a distance of $x$ (Eq. (8)), as shown in Supplementary Fig. 2.

$$x = \frac{1}{2}\mathbf{a}t_1^2 = \frac{\sqrt{6} - \sqrt{2}}{2}r \tag{8}$$

When the time consumed by water vaporization $t_0$ is less than the time consumed by water replenishment $t_1$ (Eq. (9)), the cutting of water can be realized. Under this condition, the vaporization time ($t_0$), replenishment time ($t_1$), and thickness of water ($h$) satisfy Eq. (10), from which $h$ is positively correlated with $r$ and $q$ (Eq. (11)).

$$t_1 = \sqrt{\frac{2x}{\mathbf{a}}} = \sqrt{\frac{\left(\sqrt{6} - \sqrt{2}\right)r^2}{\mathbf{g}h}} \tag{9}$$

$$t_1 > t_0 \geq h \frac{c\rho\Delta T + \rho L_V}{q} \tag{10}$$

$$h < \sqrt[3]{\frac{\left(\sqrt{6} - \sqrt{2}\right)r^2 q^2}{\mathbf{g}(c\rho\Delta T + \rho L_v)^2}} \tag{11}$$

Based on the above theoretical analysis, it can be inferred that the thickness of NEWP is an important parameter that affects the laser cutting procedure through influencing variables such as the time needed for evaporation of water, the required laser power setting, and the radius of the incision created by LCNEWP.

Second, to simulate the LCNEWP process, we analyzed the heat transfer and shape evolution of a NEWP during LCNEWP using finite element analysis (Fig. 2c, d, Supplementary Movies 1–2). Upon laser exposure, $SiO_2$ NPs around the laser irradiation point absorb laser energy, and their temperature gradually increases to over one thousand Celsius degrees (Fig. 2c). The $SiO_2$ NPs then transfer this heat to water, causing an increase in water temperature in the heated area. As the temperature of water reaches the boiling point, 100 °C, the water gradually vaporizes, and the corresponding NEWP region becomes thinner (Fig. 2d). Under the condition of constant laser power, the time required to cut a NEWP with a water thickness of 1.00 mm is about tenfold that for a NEWP with a water thickness of 0.25 mm. This is because a wider range of temperature and shape changes are generated within water of 1.00 mm thickness due to the longer required time for heat transfer, supporting the conclusion that thickness of water is a significant influencing parameter in the LCNEWP process.

Third, to confirm the relationship between NEWP thickness and the required laser power analyzed by the theoretical model, NEWPs of different thicknesses were fabricated through gradually withdrawing different volumes of water and subjected to laser processing. The top view area of the NEWP was found to shrink by only 7% when 95% of the water was extracted, which ideally allows for a large working area for laser cutting (Fig. 2e, Supplementary Fig. 3). This property results from the thin film composed of NPs on the surface of the NEWP inhibiting the shrinking force of the enclosed water. As shown in Fig. 2f, the LCNEWP procedure readily succeeds when the thickness of the NEWP is less than or equal to 1.5 mm. As NEWP thickness increases, the energy required for LCNEWP also increases, which is consistent with the results of the theoretical analysis. However, LCNEWP is difficult to achieve for NEWPs with thicknesses of or >1.75 mm. In addition, the LCNEWP strategy is also suitable for light-absorbing liquids, which require less laser energy during laser cutting in comparison to pure water (Supplementary Fig. 4).

Fourth, to obtain optimized laser cutting results, slit width was used as an index for evaluation of laser cutting and optimization of parameters that affect machining accuracy, including NEWP thickness, cutting speed, and spot size. As shown in Fig. 2g, smaller NEWP thickness corresponds to narrower slits produced during LCNEWP and thereby higher machining accuracy. This correlation between slit width and NEWP thickness holds because thinner NEWPs can be cut in a shorter period of time and with less laser energy, which minimizes the lateral transfer of heat in water and causes a smaller area of water to be vaporized. Given the above relationship, NEWPs with an optimized thickness of 0.1 mm were used for subsequent experiments. It was also found that during the LCNEWP process, excessive laser energy would heat the region of the Petri dish under the newly cut NEWP slit. While the effect of laser on the NEWP is relatively small due to the slit width is larger than the spot size. Therefore, as shown in Fig. 2g, under the condition that LCNEWP can be realized, the cutting speed did not significantly impact machining accuracy within a certain range. To evaluate the effect of laser spot size on NEWP slit width, it was determined that a laser with spot diameter 200 μm produces a laser-machined straight line with slit width 350 μm (Fig. 2g), while a smaller laser spot size produces a NEWP slit width of 200 μm (Supplementary Fig. 5). As such, NEWP thickness and spot size are key parameters that affect LCNEWP resolution.

## Various self-supporting chips fabricated through laser cutting of nanoparticle-encased water pancakes

In view of the digital control capabilities, simple operation, and rapid processing of laser cutting technology, the LCNEWP methodology can

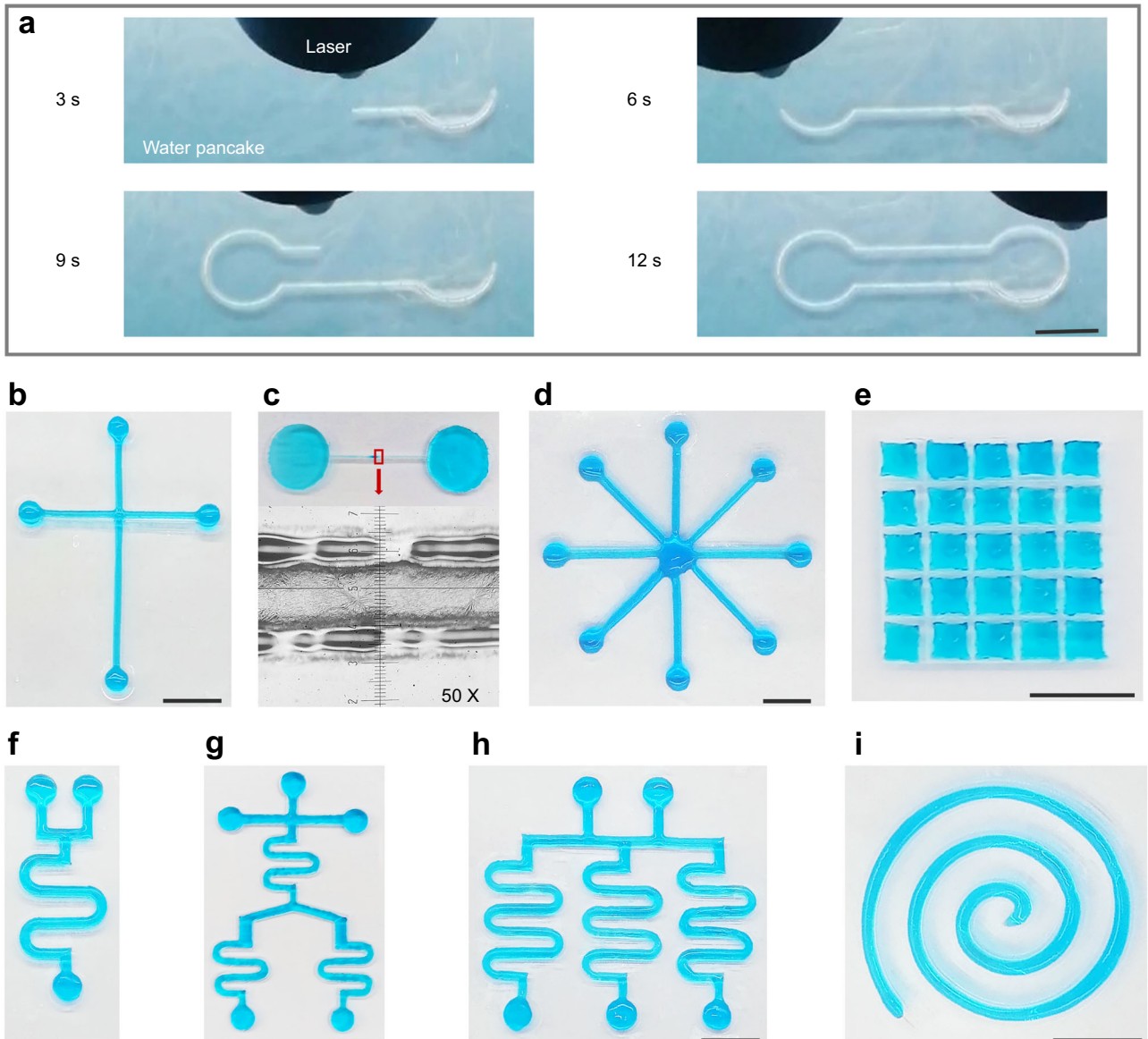

**Fig. 3 | Various SSCs fabricated through laser cutting of nanoparticle-encased water pancakes. a** Dynamic process of the SSC fabrication through laser cutting of nanoparticle-encased water pancakes. **b** Cross-type SSC chip. **c** A partial enlarged view of a straight channel and a microscope eyepiece scale. The diameter of the straight channel enlarged by 50 folds is about 1.75 cm. **d** SSC with radial array fluid channels. **e** Droplet array SSC. **f** SSC with a single curved fluid channel. **g** SSC with multiple fluid channels. **h** SSC with array of curved fluid channels. **i** Spiral SSC. The scale bars represent 1 cm.

be applied as a high-throughput strategy for machining water in NEWP. To demonstrate the aforementioned advantages, a computer-controlled laser cutting procedure for machining water in NEWP, which generates a closed-loop pattern that successfully isolated the water inside the incision in 12 s, is shown in Fig. 3a and Supplementary Movie 3. To demonstrate the capability of laser cutting as a general method for machining water in NEWP, a series of commonly used water patterns were processed. Figure 3b shows the fabricated criss-cross pattern, with very smooth, straight channel arms and a structurally stable connecting part. The resolution of the formed straight channel reaches 350 μm (Fig. 3c), similar to that of the open microfluidic chips constructed using liquid walls[14] and superior to that of mechanical shaping of liquid plasticine/pancake in previous reports[15,16]. Even the resolution of the SSCs made by laser cutting is still lower than that of the conventional solid-state microfluidic chips, it can be further enhanced through using laser with the smaller spot and reducing the fluidity of water. More complex water patterns, including

radial arrays (Fig. 3d), droplet arrays (Fig. 3e), and curved structures such as a single set of curved channel shapes (Fig. 3f), multiple sets of curved channel shapes (Fig. 3g, h), and helical channel shapes (Fig. 3i), were also successfully fabricated using laser cutting, with smooth channels being obtained. The above results support the ability of laser cutting as a general strategy to construct a variety of complex water patterns with high precision.

Through forming patterns consisting of NP films and encapsulated water, various SSCs with stable shapes in open environments without solid walls were created, presenting no obvious deformation during transfer with the substrate (Supplementary Fig. 6, Supplementary Movie 4). Such self-supporting shaped liquid entities boast many unique properties, including openness, transparency, and air permeability[17,18]. Given these properties, SSCs can successfully realize fluid manipulation (Supplementary Fig. 7), colorimetric and fluorescent detection (Supplementary Fig. 8), and applications involving gas monitoring (Supplementary Fig. 9). In addition, the issue of bubble

that normally generated in closed microfluidic chips can be easily overcome using the open solid wall-free microfluidic chips[14]. Open microfluidic chips also avoid the high shearing force found in closed microfluidic chips during fluid manipulation that can damage cells, proteins and antibodies[19]. In addition, open SSCs have the ability to isolate their internal liquid from particulate contaminants in the surrounding environment through the encapsulating NP film.

## Liquid manipulation in self-supporting chips

Manipulations of continuous liquids[20] and droplets[21–23] have attracted considerable research interest, due to their significance in biomedical, materials, chemistry, environment, and food applications. To verify the feasibility of manipulating liquids in SSCs fabricated by laser cutting, a variety of continuous liquid and droplet manipulations are demonstrated as follows.

Pumps and valves are the key components for liquid control in fluidic channels. To demonstrate the functionality of pumps and valves, a SSC consisting of two reservoirs and a straight channel was fabricated through laser cutting. As shown in Fig. 4a and Supplementary Movie 5, liquid pumping was accomplished in the manufactured SSC using syringe pumping. Initially, the SSC was connected to a syringe pump using a tygon hose (1# of Fig. 4a). Red liquid then successfully flowed from the injection reservoir to the second reservoir, with the dynamic process of syringe pump-driven liquid flow shown in (2#−6#) of Fig. 4a. However, the pump is not essential for SSCs since the flow of liquid can be driven by the pressure provided by the gravity of liquid (Supplementary Fig. 10, Supplementary Movie 6). The valve function was realized through exploiting the SSC's properties of openness, cuttability, and connectivity, which is not a complex valve structure as that of a closed microfluidic chip. Initially, the channel was cut at the channel inlet of the SSC through separating the particulate-enclosed liquid (1# of Fig. 4b). Driven by the syringe pump, the pumped liquid in the injection reservoir was prevented from flowing down the channel (2# and 3# of Fig. 4b, Supplementary Movie 7), creating a "closed" valve. After the isolated channel segments were connected using a droplet (1# of Fig. 4c), the valve opened and the red liquid was able to be successfully pumped to the other end of the channel (2# and 3# of Fig. 4c, Supplementary Movie 8). More complex and commonly used liquid manipulations in the SSC with various structures, including mixing of multiple liquids (Fig. 4d, e, Supplementary Movies 9, 10), solution concentration gradient generation (Fig. 4f, g, Supplementary Movies 11, 12), segmented solution generation (Fig. 4h, Supplementary Movie 13), and droplet array fabrication and manipulation (Fig. 4i, j), have also been successfully demonstrated in SSCs with various structures. Furthermore, the SSCs can be reused after performing liquid manipulation experiments (Supplementary Fig. 11). These liquid manipulation results demonstrate the potential of the SSCs processed by laser cutting as a general-purpose liquid manipulation platform to meet various application needs in materials, chemistry, and biomedicine fields.

## Self-supporting chips for chemical synthesis and biochemical analysis

Miniaturized chemical reactions have attracted significant attention in recent years due to their unique advantages of minimizing reagent volumes and allowing more safe, rapid, and environmentally-friendly reactions[24]. Microfluidic chips are common platforms for miniaturized chemical reactions[25]. However, the processing of traditional microfluidic chips is complicated as channels are usually closed, causing reaction product collection to be cumbersome. SSCs are easy to build, have optimal light transmittance and openness, and can be flexibly designed, making them strong choices as platforms for miniaturized chemical synthesis.

Rapid screening of optimized reaction conditions by high-throughput miniaturized chemical reactions is an effective strategy for reducing time and reagent costs in chemical synthesis. We explored the application of SSCs to an example of a high-throughput chemical reaction, cupro-ammonium complexation, using an array-type SSC. As illustrated in Fig. 5a, after ammonia water and water were introduced into the SSC from two inlets, the micro-reactors held differing ratios of ammonia water and water depending on their distances from the initial injection reservoirs. After addition of copper sulfate solution, different ratios of the reaction products were generated in the micro-reactors. In the micro-reactors that contained a low proportion of ammonia water, the reaction product consisted mainly of light blue copper hydroxide flocs. As the ratio of ammonia water in the micro-reactors increased, the reaction product in the micro-reactors became darker in blue color, signaling the gradual change of the reaction product into a dark blue copper ammine complex. This result indicates that array-type SSCs fabricated by laser cutting serve as efficient platforms for screening of reaction conditions.

For the small-volume reactant systems encountered in miniaturized chemical reactions, especially systems that are high-viscosity, efficient mixing between the reactants is of great importance for high reaction efficiency[26]. We demonstrate the application of a fabricated SSC containing curved microchannels to facilitate reactant mixing in chemical synthesis reactions. As shown in Supplementary Fig. 12, two dyes (red and green) formulated in 80% (v/v) glycerol, a highly viscous solvent, were injected into the SSC with curved microchannels and mixed efficiently. To confirm the SSC's applicability in real synthetic reactions, glycine and ninhydrin were then introduced into the SSC and reacted to generate a blue compound through isolating the mixed reactants in the micro-reactor from the fluidic channel using the valve function and reacting them independently (Fig. 5b). These observations confirm that the developed SSCs with curved microchannels efficiently promote the mixing of a small amount of reactants. Furthermore, due to SSC openness, the reaction products in the microreactors can be conveniently collected at any time for subsequent identification and application, which is important for chemical synthesis research.

As a miniaturized analytical platform, microfluidic chips have attracted significant attention in the fields of biomedicine and environment due to their advantageous properties of multifunctional integration and low reagent consumption[27,28]. However, the wider application of traditional microfluidic chips is limited by complex processing steps and high cost, making laser cutting fabricated SSCs an ideal alternative. Here, we conceptually validate the efficacy of SSCs as analytical platforms through application to the simultaneous colorimetric sensing of multiple metal ions (Fig. 5c) and biochemical indicators (Fig. 5d). Due to the good light transmittance of the SSC, the colorimetric sensing signal was easily collected. We further developed a SSC, consisting of an array of droplets covered NPs, which was integrated with synthetic biosensors to detect nucleic acids (Fig. 5e). Although the NPs coated on the SSC were breathable and water permeable, they provided some blocking effect against solid interferents in the external environment relative to exposed droplets, which was beneficial for the operation of synthetic biosensors. After incubation with nucleic acid sensors, the droplet arrays containing a range of nucleic acid concentrations were colored shades of yellow (Fig. 5f). Further colorimetric analysis (Fig. 5g) determined a positive correlation between nucleic acid concentration and grayscale droplet signal intensity, indicating the successful function of SSCs as the nucleic acid biosensors. And the droplet array also allowed data from sixteen samples to be collected in one experiment, which was conducive to the realization of high-throughput nucleic acid detection. These findings demonstrate that a variety of SSCs structures can be flexibly designed through laser-cutting as analytical platforms for detections of metal ions and biomarkers. Given the ease of their processing and detection of optical signals, SSCs also hold potential for development in colorimetric, fluorescent and chemiluminescent sensing fields.

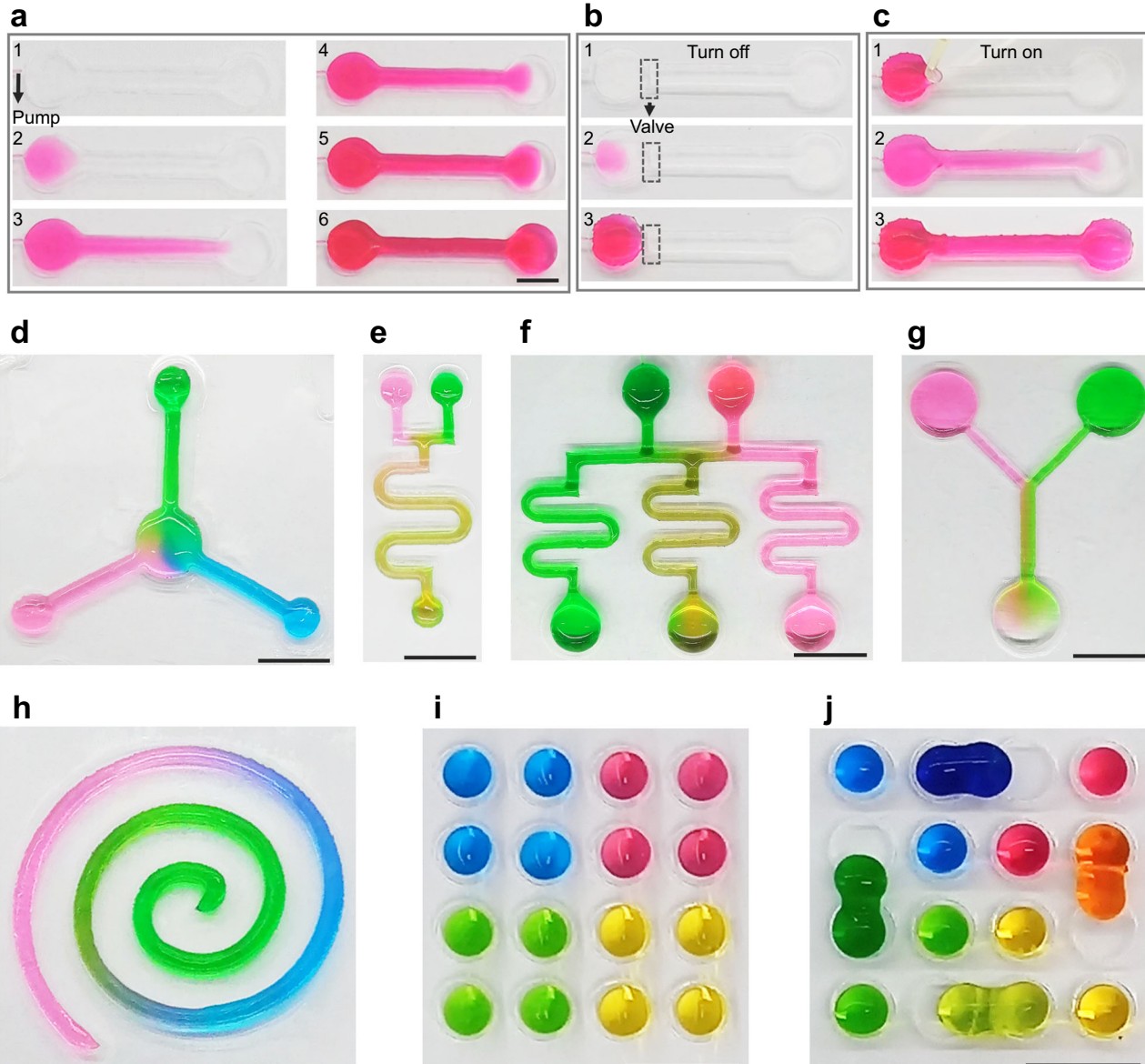

**Fig. 4 | Liquid manipulation within the fabricated SSCs. a** Liquid pumping.
**b** Turning off the valve by cutting the fluid channel. **c** Turning on the valve by
connecting the fluid channel. **d** Mixing of various liquids (blue, green and red
liquids) in radial-array fluid channels. **e** Mixing of two liquids (green and red liquids)
in a single curved fluid channel. **f** Concentration gradient generated in an array of
curved fluid channels (green and red liquids). **g** Concentration gradient generated
in a "Y" fluid channel (green and red liquids). **h** Segmented solution generated in a
spiral SSC. **i** Dispersion of different liquids in droplet arrays. **j** Liquid mixing through
droplet combination. The scale bars represent 1 cm.

## Self-supporting chips for liquid metal manipulation and conductive hydrogel synthesis

As fascinating materials with both metallic and liquid properties,
liquid metals have found applications in material synthesis, catalysis,
flexible electronics, and drug delivery fields[29,30]. Manipulations of
liquid metals using magnetic[31], electric[32], optical[33,34] and other meth-
ods have also been the focus of a growing body of research in
robotics. Such applications thus necessitate the development of
platforms for intelligent manipulation of liquid metal in complex
structures. To explore the feasibility of SSCs in electrokinetic
manipulation of liquid metals, we developed a maze-shaped SSC with
three different paths filled with NaOH solution and containing a small
liquid metal sphere (gallium-indium eutectic). When the SSC was
connected to a direct current (DC) circuit, the liquid metal sphere was
found to migrate from the cathode to the anode by autonomously
choosing a feasible circuit from the three paths. When the electrode

position was changed, the liquid metal sphere, driven by the electric
field, was able to alter its movement to select the new correct path
from the three channels. Due to the transparency of the SSC, the
dynamic process of the liquid metal sphere moving inside the SSC
could be recorded in real time, as shown in Fig. 6a and Supplementary
Movies 14 and 15. We also explored the mechanisms underlying the
electrokinetic manipulation of liquid metal (gallium-indium eutectic)
in the SSC. As illustrated in Fig. 6b and Supplementary Fig. 13a, when
the SSC was filled with NaOH solution, a reaction occurred between
Ga, present in the liquid metal alloy, and $OH^-$ ions from NaOH[35].
Through the reaction, the liquid metal gained surface negative charge
and achieved electric field-driven movement from the cathode to the
anode. In contrast, when submerged in an acidic solution such as HCl,
the liquid metal reacted with $H^+$, providing surface positive charge
that promoted electromigration from anode to cathode (Fig. 6c,
Supplementary Fig. 13b). This exploration elucidates the efficacy of

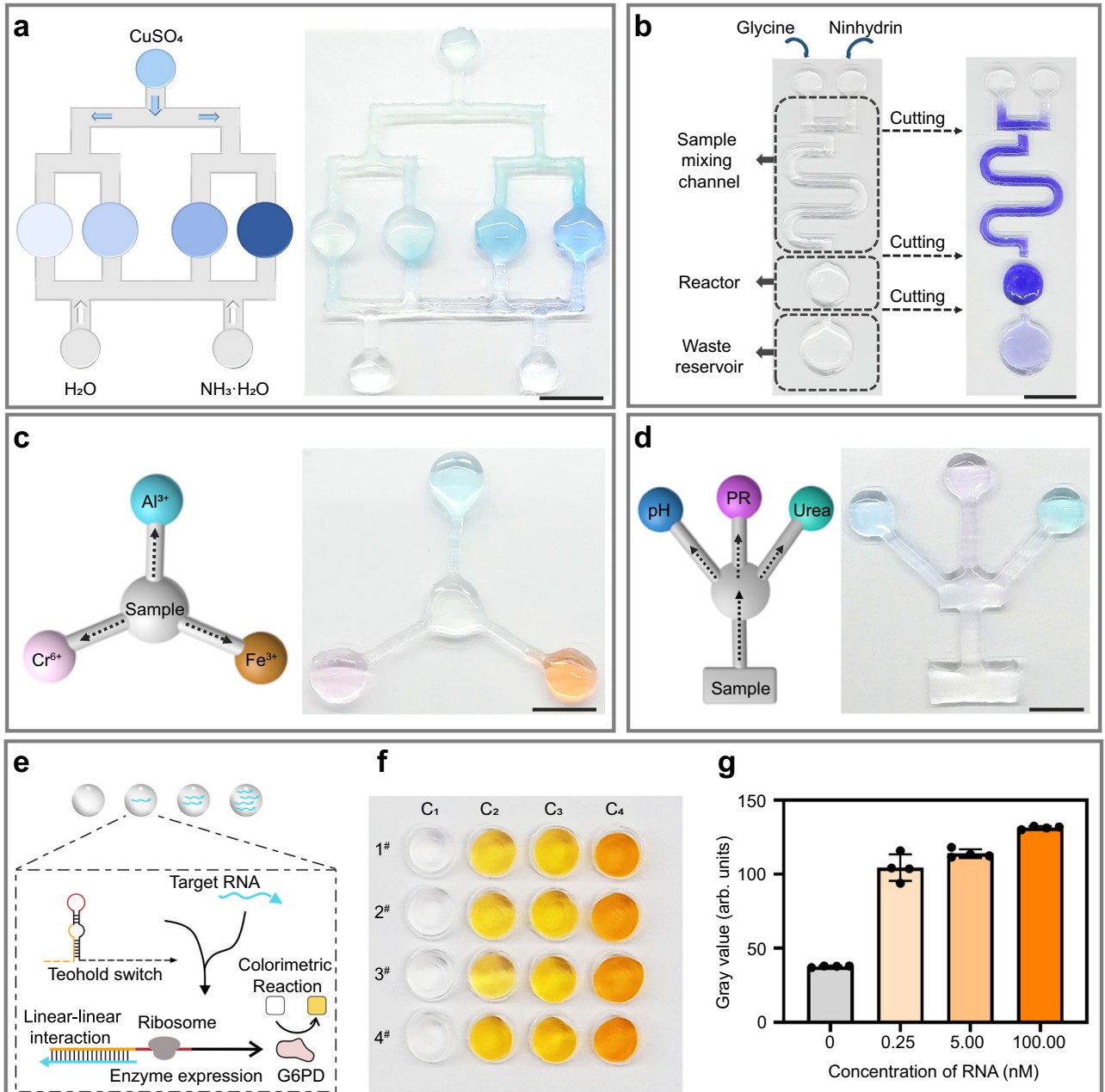

**Fig. 5 | Chemical synthesis and biochemical sensing within SSCs. a** Schematic diagram and experimental results of the cupramine complexation reaction. **b** Schematic diagram and photograph of the synthesis reaction of glycine and ninhydrin. **c** Schematic diagram and experimental results of simultaneous detection of metal ions: $Fe^{3+}$ using the phenanthroline method, $Al^{3+}$ using the chrome azure S method, and $Cr^{6+}$ using the diphenylcarbazide method. **d** Schematic diagram and experimental results of simultaneous detection of biochemical indicators: protein (PR) detection using the biuret method, urea detection using a

chromogenic kit, and pH detection using bromothymol blue. **e** Schematic illustration of nucleic acid detection using synthetic biosensors and glucose 6-phosphate dehydrogenase (G6PD) colorimetric reaction. **f** Results of four parallel nucleic acid detection tests in a droplet array containing solutions with different RNA concentrations (0, 0.25 nM, 5.00 nM, 100.00 nM). **g** Colorimetric analysis of each droplet in the droplet array. Data of distinct samples are presented as mean ± s.d., $n = 4$. The scale bars represent 1 cm.

SSCs as open, transparent, and flexibly designed platforms for intelligent manipulation of liquid metals.

Hydrogels, multifunctional hydrophilic polymeric materials of great interest, have been widely applied in tissue engineering[36], regenerative medicine[37], drug delivery[38], biochemical sensing[39], and electronics[40]. In particular, conductive hydrogels have attracted extensive attention in the flexible electronics field. Hydrogel patterning plays a vital role in the development of multifunctional hydrogels[41–43]. The good transparency, openness and channel design flexibility of SSCs make them ideally equipped as platform for the

synthesis of patterned hydrogels. As a concept presentation, we used SSCs as molds for the synthesis of patterned conductive hydrogels in the form of a man and woman. As shown in Fig. 6d, e, patterned SSCs fabricated by laser cutting were filled with a precursor solution of polyacrylamide hydrogel that also contained conductive $Na^+$ and $Cl^-$ ions. Synthesis of the patterned conductive hydrogels was completed after 1 h of room temperature incubation (Fig. 6f), after which the hydrogels were used as part of a DC circuit to successfully light up an LED (Fig. 6g, Supplementary Movie 16), validating their electrical conductivity. The result supports the role of SSCs as a reliable platform

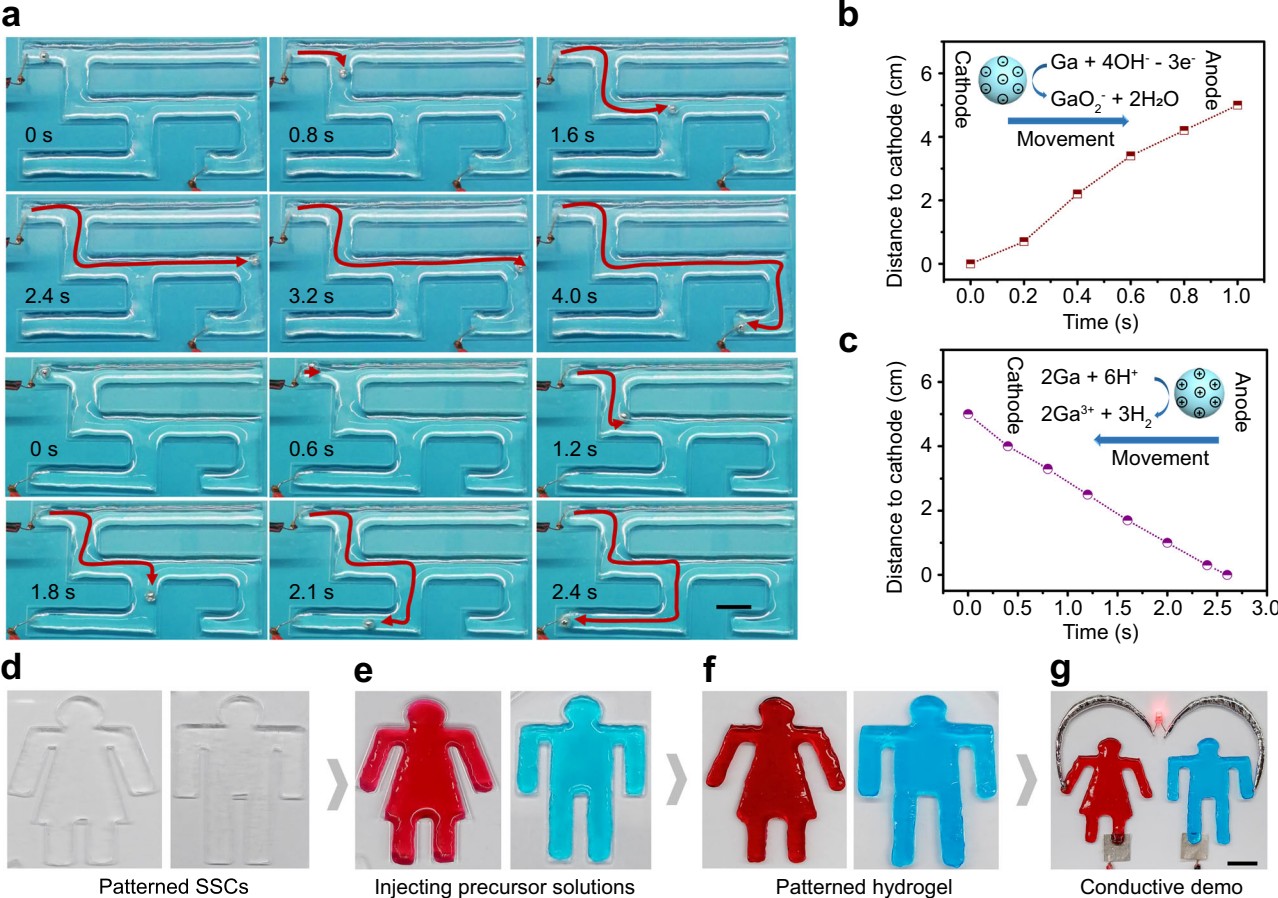

**Fig. 6 | Liquid metal manipulation and conductive hydrogel synthesis within self-supporting chips (SSCs). a** Dynamic illustration of electrokinetic manipulation of liquid metal in the SSC. Experimental condition: 100 mM NaOH solution injected into the SSCs; 80 V applied DC voltage. The scale bar represents 1 cm. **b** Calibration curve of the distance electromigrated by the liquid metal from cathode to anode in an alkaline solution over time. The graph inset displays a schematic diagram of the migration of liquid metal from the cathode to the anode. Experimental condition: 50 mM NaOH solution injected into the linear channel; 50 V applied DC voltage. **c** Calibration curve of the distance electromigrated by the liquid metal from anode to cathode in an acidic solution over time. The inset is a schematic diagram of the migration of liquid metal from the anode to the cathode. Experimental condition: 20 mM HCl solution injected into the linear channel; 50 V applied DC voltage. **d** SSCs processed by laser cutting. **e** SSCs injected with hydrogel precursor solutions. **f** The patterned conductive polyacrylamide hydrogels synthesized in SSCs. **g** Conductivity demonstration of synthesized patterned hydrogel. The scale bar represents 1 cm.

for synthesis of patterned hydrogels with the unique advantages of transparency and openness for photocrosslinking and exfoliation of hydrogel synthesis.

## Self-supporting chips for cell culture and drug screening
Due to their low reagent consumption and flexibility of design, microfluidics technologies have been widely explored for cell culture and drug screening applications[44,45]. Concentration gradient generation and gas permeability are critical properties in microfluidic chips for high-throughput screening[46]. As such, given their good gas permeability, transparency and openness, SSCs can serve as effective platforms for cell culture-based drug screening. As a concept presentation, a SSC for concentration gradient generation, consisting of gradient dilution channels, cell culture reservoirs, and liquid waste reservoirs, was fabricated through laser cutting (Fig. 7a).

To verify the feasibility of the proposed SSC as a platform for cell culture and drug screening applications, we selected MDA-MB-231 breast cancer cells as a cancer cell model and doxorubicin hydrochloride (DOX·HCl) as an anticancer drug representative and explored the cytotoxic effect of different concentrations of DOX·HCl on MDA-MB-231 cells. As shown in Fig. 7b, after MDA-MB-231 cells were injected into the three cell culture chambers and adhered to the chamber walls,

they were exposed to DOX·HCl along a concentration gradient that was generated through serial DOX·HCl dilution with cell culture medium over the array of curved fluid channels. The concentration of DOX·HCl decreased from the first (No. 1) to the third cell culture chamber (No. 3). To analyze the cytotoxic effect of three DOX·HCl solutions of differing concentrations, cell counting kit-8 (CCK8) staining and fluorescent staining methods were used to detect MDA-MB-231 cell viability after drug application. The CCK8 staining results in Fig. 7c indicate that cell viability decreased with increasing DOX·HCl concentration. Fluorescent staining results in Fig. 7d similarly showed a correlation between higher DOX·HCl concentration and a greater number of dead cells (red) and lesser number of viable cells (green). The cell counting results in Fig. 7e were consistent with those of CCK8 staining, confirming this cell viability-DOX·HCl concentration relationship. Therefore, given their flexible design, simple processing, superior gas permeability, and transparency in comparison to traditional cell culture microchips, SSCs hold promising potential as platforms for applications in cell culture-based biomedicine.

## Discussion
This manuscript introduces a strategy that machines water through laser cutting NEWPs, water coated with $SiO_2$ NPs, with submillimeter

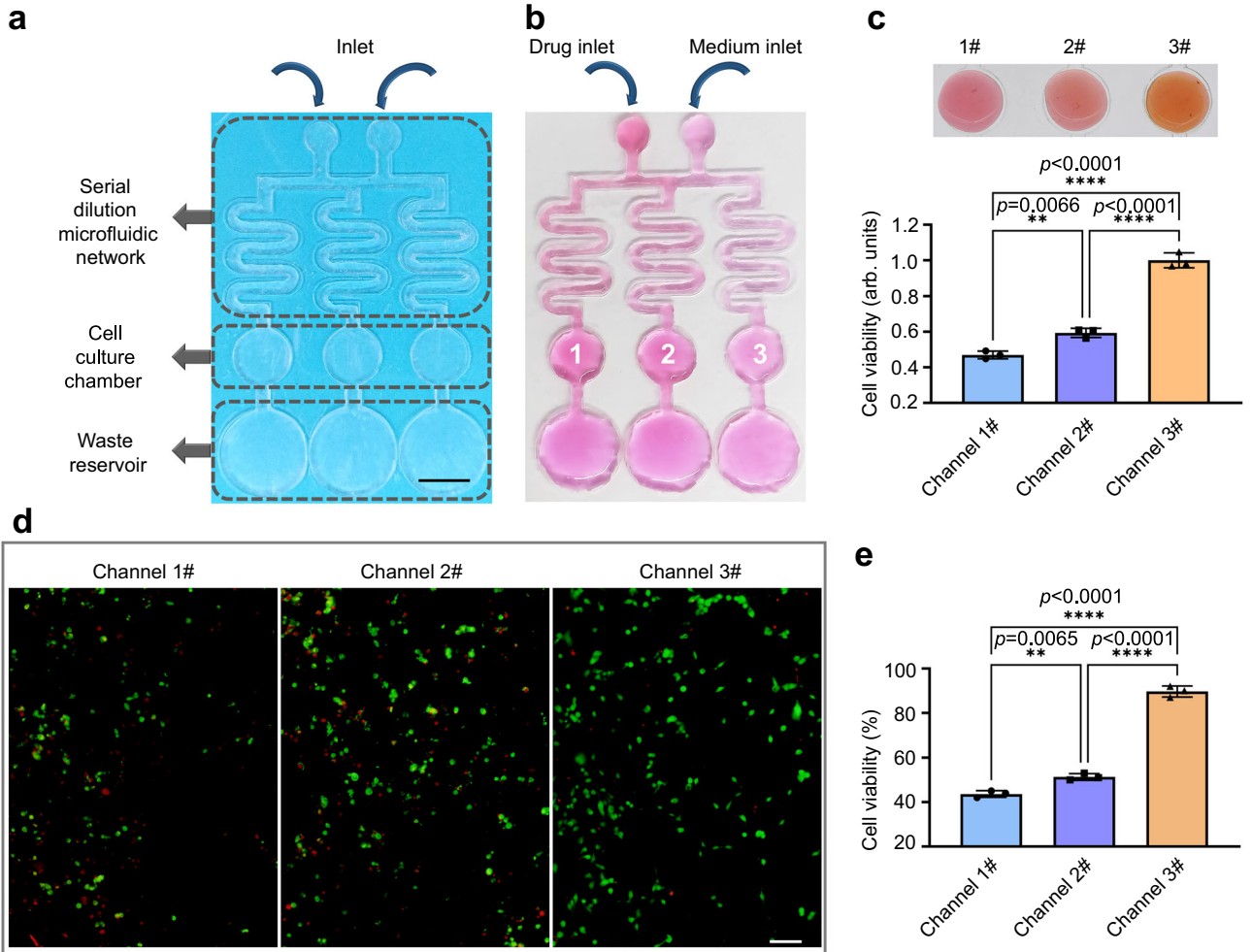

**Fig. 7 | Drug screening within self-supporting chips (SSCs). a** Schematic diagram of the SSC structure. The scale bar represents 1 cm. **b** Gradient dilution of DOX·HCl in the SSC using culture medium. **c**, CCK8-stained MDA-MB-231 cells in different cell culture chambers after serial dilution of DOX·HCl. A brighter orange color cell culture stain indicates higher cell viability. The histogram depicts the results of quantitative analysis of CCK8 staining results using spectrophotometry. Data of distinct samples are presented as mean ± s.d., $n = 3$. **d** Images of fluorescence-stained MDA-MB-231 cells in different cell culture chambers after serial dilution of DOX·HCl, in which the red and green dots respectively represent dead and live cells. The scale bar represents 100 μm. **e** The histogram depicts the results of counting analysis of fluorescent staining. Data of distinct samples are presented as mean ± s.d., $n = 3$.

thickness. This strategy resolves the challenges of precisely machining water through laser cutting by minimizing the effect of properties of water, achieving the development of patterned and self-supporting water shapes in NEWP. Theoretical analysis, numerical simulation, experimental optimizations, and processing of various water patterns strongly confirm the efficacy and feasibility of the developed method, supporting the work's contribution to overcoming the difficulties of water laser cutting. In addition, the given analysis, numerical simulation, and experimental study of the LCNEWP process and cutting accuracy parameters can provide inspiration for researchers exploring the processing of other liquid materials. In practical application, patterned water processed through laser cutting can form diverse open, transparent, breathable, and flow control SSCs. As verified through chemical synthesis, biochemical sensing, liquid metal manipulation, conductive hydrogel synthesis, and drug screening demonstrations, such SSCs are ideal for numerous applications across chemistry, health, material science, and biology.

The laser machine used in our work can achieve a LCNEWP fabrication resolution as low as 200 μm. As SSCs are formed by self-supported liquid, it is difficult for SSCs to maintain vertical shape stability due to the effect of gravity. As such, the processing accuracy and 3D construction capability of the SSCs proposed in our work are not comparable to those of conventional microfluidic chips. However, the accuracy of SSCs may be improved through utilizing lasers of smaller spot sizes and reducing the mobility of water, proposals which will be explored in our future study. Moreover, machining of water in NEWPs through laser cutting, which holds the advantages of flexible design, precision, and fast machining, has application potential in the industrial processing of personalized microfluidic chips. We thus anticipate opportunities for cooperation with teams in physics, biology, medicine, chemistry, materials, and other interdisciplinary fields to promote diverse applications of the proposed strategy.

## Methods

### Chemicals

Copper sulfate ($CuSO_4$, 99.0%), ferric chloride ($FeCl_3$, 99.0%), aluminum chloride ($AlCl_3$, 99.0%), sodium chromate ($Na_2CrO_4$, 99.0%), sodium chloride (NaCl, 99.5%), sodium hydroxide (NaOH, 96.0%), bromothymol blue (95.0%), amaranth (85.0%), brilliant blue (85.0%), tartrazine (85.0%), ninhydrin (98.0%), and ammonium persulfate (98.0%) were all purchased from Aladdin Reagent Co., Ltd. (Shanghai, China). Ammonia ($NH_3 \cdot H_2O$, 25%), hydrochloric acid (HCl, 36%), N,N,N',N'-tetramethylethylenediamine (TEMED, 98.0%), and glycerol (99.0%) were purchased from Sinopharm Chemical Reagent Co., Ltd

(Beijing, China). Acrylamide (99.0%), N, N′-methylenebisacrylamide (99.0%), glycine (99.0%), and liquid metal (eutectic gallium-indium, EGaIn) were purchased from Sigma Aldrich (USA). Sodium copper chlorophyll (95.0%), ethanol (EtOH, 99.8%), tetraethyl orthosilicate (TEOS, 95.0%), and hexamethyldisilazane (HMDS, 99.0%) were purchased from Meryer Chemical Technology Co., Ltd. (Shanghai, China). $Fe^{3+}$, $Al^{3+}$, $Cr^{6+}$, and urea detection kits and biuret reagent were purchased from Luheng Environmental Technology Co., Ltd. (Hangzhou, China). A PURExpress in vitro protein synthesis kit was purchased from New England Biolabs (USA). RNase inhibitor was purchased from F. Hoffmann-La Roche Ltd. (Switzerland). A target RNA sequence for RNA detection was synthesized by Sangon Biotech Co., Ltd. (Shanghai, China). A glucose 6-phosphate dehydrogenase (G6PD) detection kit was purchased from Beyotime Biotechnology Co., Ltd. (Shanghai, China). Bovine serum albumin and urea were purchased from Solarbio Life Sciences Co., Ltd. (Beijing, China). Dulbecco's modified eagle medium with high glucose (DMEM), penicillin/streptomycin (P/S), and trypsin were obtained from Gibco Life Technologies Corp. (USA). Fetal bovine serum (FBS) and phosphate buffered saline (PBS) were purchased from InCellGene LLC. (USA). Doxorubicin hydrochloride was purchased from Selleck Chemicals LLC. (USA). A Cell Counting Kit-8 was obtained from Dojindo Laboratories (Japan), and a Live/Dead Viability/Cytotoxicity Kit was purchased from Thermo Fisher Scientific Inc. (Shanghai, China).

### Preparation of silica sol
Silica sol was prepared using the following steps: (1) mixing 50.00 mL of EtOH with 5.00 mL of TEOS and stirring for 15 min; (2) adding 1.80 mL of ammonia to the mixed solution and stirring for 1 h; (3) aging at room temperature for 7 days; (4) adding 2.82 mL of HMDS to the solution and stirring for 2 h; (5) aging at room temperature for 1 h; and (6) adding 100.00 mL of EtOH to the solution.

### Preparation of NEWP
The NEWP was prepared using the following steps: (1) adding 10.00 mL of 10 mg mL$^{-1}$ $SiO_2$ NPs sol to a 15 cm × 15 cm Petri dish; (2) shaking the Petri dish to coat the dish inner surface with $SiO_2$ NPs sol; (3) removing and recovering excess $SiO_2$ NPs sol; (4) leaving the Petri dish standing at room temperature for 15 min; (5) adding 45 mL of water drop by drop to the Petri dish using a syringe; (6) adhering hydrophobic $SiO_2$ NPs with diameter of around 20 nm (Supplementary Fig. 14) to the droplet surface from the Petri dish; (7) forming a water pancake encased in hydrophobic $SiO_2$ NPs through moving the droplets to fuse with one other; and (8) removing 42.75 mL of water from the water pancake.

### Operating procedures for fabrication of SSCs through LCNEWP
The CorelDRAW drawing software was used to design a line pattern of a chip boundary. SSCs containing a small amount of liquid were obtained through directly engraving the line pattern onto a thin NEWP using a continuous-wave (CW) laser of wavelength 10.6 μm, produced by a laser engraving machine (EMT-9060, EMIT laser, China). The laser cutting machine was equipped with a focusing lens with focal length 63.5 mm, which generated a laser spot with a diameter of around 0.2 mm. The laser had a scanning speed of 10 mm s$^{-1}$ and power density of 210 J mm$^{-2}$ during the NEWP cutting process.

### Finite element analysis on evolution of water shape and temperature distribution during laser cutting
Simulations of the evolution of water shape and temperature distribution during laser cutting were conducted using COMSOL Multiphysics 6.0 software (COMSOL, Sweden) (Fig. 2c, d). In the finite element analysis procedure, the NEWP was divided into two parts, i.e., the $SiO_2$ NP film and water. First, the temperature distribution of the $SiO_2$ NP film was simulated using the solid heat transfer method, in

which the derived temperature distribution of the $SiO_2$ NPs coming in contact with the upper water surface was fitted into a Gauss function. This is shown in Eq. (12).

$$T(x) = T_0 + A*\exp\left(-\frac{2x^2}{\omega^2}\right) \qquad (12)$$

where $T$ and $x$ refer to the temperature and position of the $SiO_2$ NPs, respectively. The parameters $T_0$, $A$ and $\omega$ in Eq. (12) vary with time $t$. The fitted results are shown in Eq. (13).

$$\begin{cases} T_0(t) = -41 + 61*\exp\left(-\frac{25t}{18}\right) \\ A(t) = 1938 - 1938*\exp\left(-\frac{5t}{2}\right) \\ \omega(t) = 2.95 - 0.95*\exp(-2t) \end{cases} \qquad (13)$$

Second, the temperature of the upper surface of the water was set equal to the temperature of the underlying $SiO_2$ NPs, as both elements are directly in contact with each other. The temperature distribution and shape evolution of water in NEWPs with thicknesses of 0.25 mm and 1.00 mm were simulated by establishing a fluid heat transfer module and using a sensible enthalpy method, respectively.

### Manipulating liquids in SSCs
The SSCs used in the fluid manipulation demonstrations (Fig. 4a–j) were all manufactured through LCNEWP. Blue, green, red, and yellow liquids, respectively solutions of brilliant blue, sodium copper chlorophyll, amaranth, and tartrazine with concentrations of 0.5 mg mL$^{-1}$, were utilized for fluid manipulation. A syringe pump (TS-2A syringe pump, Longer Precision Pump Co., Ltd., China) provided the driving force for the flow of liquids, controlling the velocity of all injected liquids at 100 μL min$^{-1}$. During syringe-pumping of liquid into the SSCs, the tygon hose connected to the syringe pump was not inserted inside the SSCs, but instead fixed near their injection ports (Supplementary Fig. 15). The results of all liquid manipulations were recorded using a camera.

As shown in Fig. 4a, a syringe pump was connected to an SSC containing two reservoirs connected by a linear channel and used to deliver red liquid. As shown in Fig. 4b, an identical SSC, with two reservoirs and a fluidic channel, was used to demonstrate the valve function. The left reservoir of the SSC was first disconnected from the linear channel and filled with red liquid using a syringe pump. The reservoir and channel were then reconnected using droplets to "open" the valve and enable pump-facilitated driving of liquid flow (Fig. 4c). As shown in Fig. 4d, an SSC consisting of three injection reservoirs and a reaction cell was fabricated. Red, green, and blue liquids were respectively injected at the same injection speed into the three injection reservoirs using syringe pumps. As shown in Fig. 4e, an SSC consisting of two injection reservoirs, a curved channel, and a reaction cell was constructed. A syringe pump was used to simultaneously inject red and green liquids from two injection reservoirs at the same flow rate. As shown in Fig. 4f, g, two SSCs, the first containing two injection reservoirs, three tortuous channels, and three reaction cells and the second containing two injection reservoirs and a Y-shaped channel, were processed. In both SSCs, red and green liquids were simultaneously injected into two injection reservoirs at the same flow rate using a syringe pump. As shown in Fig. 4h, a spiral-shaped SSC was fabricated. Red, green, and blue liquids were injected simultaneously into different injection ports of the microfluidic chip using a syringe pump. As shown in Fig. 4i, a 4×4 droplet array was fabricated and the selected droplets were manually injected with blue, green, red, or yellow liquid using a syringe. As shown in Fig. 4j, a hydrophobic plastic rod was then used to push droplets to achieve coalescence and droplet-to-droplet mixing.

## Cupro-ammonium complexation reaction in SSC

An array-type SSC with three injection ports and four reaction cells was fabricated to demonstrate the application of the SSC in the cupro-ammonium complexation reaction. Aqueous solutions containing 200 mM ammonia, deionized water, and 25 mM copper sulfate were syringe-pumped into the three injection ports at respective injection rates of 0.05, 0.05, and 0.10 mL min$^{-1}$. The duration of injection was 2.5 min. The reaction was then incubated for 5 min, and a camera was used to capture the results of the reaction (Fig. 5a).

## Chemical reaction of ninhydrin and glycine in SSC

To demonstrate the applications of SSCs as platforms for chemical synthesis, an SSC containing two injection ports, a curved channel, and a reaction cell was used to facilitate the synthesis of ninhydrin and glycine. 25 mM glycine and 50 mM ninhydrin were injected into the separated injection ports at 0.01 mL min$^{-1}$ for 2.5 min using a syringe pump; during this time, the reactants were mixed in the curved channel. After the injection, the connection between the channel and the reaction cell was cut to produce an isolated reactor. After 1 h of reaction, the results of the reaction were documented using a camera (Fig. 5b).

## Detection of metal ions using SCCs

To implement the simultaneous detection of three metal ions, a radial SSC containing one injection port and three reaction cells was fabricated. 50 μL of three chromogenic reagents respectively used in the detection of $Fe^{3+}$, $Al^{3+}$, and $Cr^{6+}$ ions were added dropwise to three reaction cells. A 500 μL sample containing 1.0 mg L$^{-1}$ $Fe^{3+}$, 0.2 mg L$^{-1}$ $Al^{3+}$, and 0.5 mg L$^{-1}$ $Cr^{6+}$ was then manually injected from the middle inlet. After 10 min of reaction, the reaction cells were photographed using a camera (Fig. 5c).

## Detection of biochemical targets in SSC

To demonstrate the simultaneous detection of three biochemical targets, a tree-shaped SSC containing one injection port and three reactors was fabricated. 50 μL of chromogenic reagents used in the detection of protein, urea, and pH were added dropwise to the three reactors. A 1.50 mL sample containing 100 mg L$^{-1}$ albumin and 10 mg L$^{-1}$ urea was then manually injected from the injection port. After 20 min of reaction, the reaction cells were photographed with a camera to acquire the reaction results (Fig. 5d).

## SSC sensor for nucleic acid detection

A SSC consisting of an array of droplets was fabricated to implement high-throughput nucleic acid detection. A synthetic biology RNA sensing system was constructed through sequentially adding cell-free enzyme expression system, RNase inhibitor (0.5%, Roche, 3335402001), and linear DNA constructs encoding toehold switch sensors and target RNA to each droplet. The cell-free enzyme expression system was performed using the PURExpress In Vitro Protein Synthesis Kit (New England Biolabs, E6800L). The target RNA consisted of a 36nt sequence (GGG UGA UGG GAC AUU CCG AUG UCC CAU CAA UAA GAG CAA GAC AAU GGU AAG UAG UAA UAG AUA AG). The toehold switch (GGG AUC UAU UAC UAC UUA CCA UUG UCU UGC UCU AUA CAG AAA CAG AGG AGA UAU AGA AUG AGA CAA UGG AAC CUG GCG GCA GCG CAA AAG AUG CGU AAA) contains a single-stranded domain, known as a toehold, at its 5' end that provides an initial reaction site for binding between target and switch RNAs. The total volume of the sensing system within each droplet was 30 μL. The droplets were immersed in a 37 °C water bath for 50 min, after which the nucleic acid detection results were obtained using the camera (Fig. 5f).

## Manipulation of liquid metal in SSCs

A labyrinth-type SSC was fabricated through laser cutting to perform manipulation of liquid metal. First, the SSC was injected with 100 mM NaOH solution. Two platinum electrodes connecting to a DC power supply were then inserted into the SSC to be in contact with the NaOH solution. A liquid metal sphere was added to the NaOH solution near the cathode in the SSC through a syringe. Finally, a DC voltage of 90 V was applied to the microfluidic chip and a camera was used to record the dynamic migration process of the liquid metal (Fig. 6a).

## Synthesis and application of patterned hydrogels in SSCs

Two SSCs in the form of a man and a woman were fabricated to demonstrate the synthesis of patterned conductive hydrogels. Initially, the patterned SSCs were each injected with 1 mL of a mixed solution of hydrogel precursor solution, cross-linking agent, and catalyst. The hydrogel precursor solution was prepared by dissolving 0.08 g acrylamide, 0.01 g methylenebisacrylamide and 0.02 g NaCl in 988 μL deionized water. The crosslinker was 2 μL of a 90 mg mL$^{-1}$ solution of ammonium persulfate. The catalyst was 10 μL of TEMED. After standing at room temperature for 30 min, patterned hydrogel synthesis was completed. Hydrogel color was modified through the addition of water-soluble pigments to the precursor liquid of the hydrogel. To collect the synthesized hydrogels for subsequent applications, deionized water was used to rinse off the nanoparticles on the surface of the hydrogels, which took only 5 s. Finally, the synthesized hydrogel was connected to liquid metal wire, a light-emitting diode (LED), and a 90 V DC power source, forming a current loop that successfully lit the LED (Fig. 6g).

## Cell culture-based drug screening in SSC

To explore the toxic effects of different concentrations of drug (DOX•HCl) on MDA-MB-231 cells, an array-type SSC with three sets of curved channels for generation of solution concentration gradient was fabricated. The deionized water enclosed in the SSC was first replaced with culture medium consisting of DMEM, 10% FBS, and 1% P/S. The three cell culture chambers in the SSC were then isolated from the channels through cutting and each inoculated with the same number of MDA-MB-231 cells. After incubating for 12 h at 37 °C in a humid environment containing 5% $CO_2$, the cell culture chambers were reconnected to the fluidic channels. To generate drug concentration gradients in the three cell culture chambers, 100 mM DOX•HCl and culture medium were both injected at a rate of 0.2 mL min$^{-1}$ into the two injection ports for 4 min. The cell culture chamber was then again disconnected from the microfluidic channel and incubated at 37 °C in a humidified environment containing 5% $CO_2$ for 24 h. Cell viability was determined through CCK8 and dead/alive staining, in which CCK8 staining and dead/alive staining reagents were each added to the three cell culture chambers. Under the CCK8 stain, the chambers were incubated for 1 h, then photographed using a camera, with a microplate reader being used to analyze the absorbance. Under the dead/alive stain, the chambers were incubated for 0.5 h and viewed under a fluorescence microscope.

## Statistical analysis

Statistical comparisons were performed with GraphPad Prism 9.2.0 using one-way analysis of variance (ANOVA) with multiple comparisons. The threshold for statistically significant differences between groups was $p < 0.05$.

## Reporting summary

Further information on research design is available in the Nature Portfolio Reporting Summary linked to this article.

## Data availability

All data in this work are available in the manuscript and in the Supplementary information section. Source data are provided with this paper.

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

## Acknowledgements

This work was financially supported by the Natural Science Foundation of Shannxi Province, China (2020JC-06, F.L.), the Key R&D Plan of Shaanxi Province (2021SF-168, F.L.), Fundamental Research Funds for the Central Universities (SY6J007, 22127803HZ, F.L.) and Projects of Interdisciplinary, NWPU (NO.0202022GH0306, X.L.).

## Author contributions

F.L., J.N. and X.L. designed the study, in consultation with F.X.; J.N., W.L., J.X.L., X.P., Y.L., C.Z., K.Y. and Y.Z. performed the experiments, collected and analyzed the data. X.L., W.L., X.P. and J.N., built the simulation model; F.L., J.N., X.L., J.X.L. and F.X. prepared the manuscript. All authors discussed the experiments, read and commented on the manuscript.

## Competing interests

The authors declare no competing interests.
