## [Peer Review File · Nature Communications]

Reviewer comments, first round

Reviewer #1 (Remarks to the Author):

This paper presents a new method for direct machining and flexible patterning of water by using laser. The developed technique was successfully applied to some relevant microfluidic applications including liquid manipulation, chemical synthesis, biochemical analysis, liquid metal manipulation, conductive hydrogel synthesis, cell culture, and drug screening. The paper is well written and well organized. The results obtained should be of great interest to those working in related fields as well as in the wider field. Therefore, I think this paper can be published in Nature Communications if the authors can properly address my comments listed below.

- (1) Laser cutting mechanism is unclear. Why can the NP film enhance the absorption. SiO₂ has absorption only in VUV (shorter than 190 nm) and IR (longer than 3 μm). What kind of laser was used? In this context, more details of laser cutting system (wavelength, pulse width or CW, focusing lens (NA), spot size, etc.) and irradiation conditions (power density or fluence, scanning speed, etc.) should be described in Methods.
- (2) Temperature of the NP film increased by laser irradiation and temperature distribution generated in water after laser irradiation should be simulated.
- (3) Details on formation of liquid pancake (Fig. 1 a1-2) should be also described in Methods.
- (4) Scale bars should be included in Fig. 1 b.
- (5) In Fig. 2 f, why is the slit width approximately equal to the water thickness regardless of scan speed? Is the slit width larger than the spot size? Can smaller spot size create narrower slit? The inset photo shows the irregularities in the slit width. What is the standard deviation of the width? Will such irregularities affect the applications?
- (6) The authors mentioned some advantages of SSC over the conventional solid microfluidic chips. What are the disadvantages of SSC? Fabrication resolution (accuracy of 350 μm) is much worse than the fabrication technique of conventional solid microfluidic chips. 3D microfluidic structures cannot be fabricated. Please comment on these.
- (7) The authors said "in terms of industry, the precise machining of water through laser cutting offers far simplified and lower material consumption pathway for patterning water compared with the traditional method of processing pre-patterned containers, which can reduce time and material costs in the field of microfluidic, especially". However, solid microfluidic chips can be currently mass-produced by the embossing technique with cheap costs. The authors should comment on this.

Reviewer #2 (Remarks to the Author):

In this study, the authors have proposed a method for laser micromachining of water coated with hydrophobic nanoparticles. They have showcased the applications of this technique by fabricating various self-supporting chips. Below are some critical issues of this study:
Firstly, the title of this manuscript is misleading. While the title looks very promising for Nature Communication journal, the main proposed technique does not offer direct laser micromachining of water. In fact, the authors have laser machined the "liquid marble" film. They refer to this liquid marble as "non-wetted liquid pancake". Also, in page 4 of the manuscript, it is mentioned that more than 95% of the enclosed water had been withdrawn from the "liquid pancake". As such, the method is no longer "directly" machining the water. Since the SiO₂ nanoparticles are transparent, it creates the illusion that the laser is cutting the water, which is scientifically not correct.
Secondly, While the proposed technique looks interesting, the advantages of this strategy are not clear. There are no added advantages to creating this substrate for laser cutting. For example, liquid manipulation in "self-supporting chips" still needs conventional valves and external pumps. So, why one should make the fabrication overcomplicated by coating the water with hydrophobic SiO₂ nanoparticles. There are many health concerns and safety hazards in handling these nanoparticles, as well. They listed the following items as the main advantages of these platforms: openness, transparency, breathability, liquid morphology and flow controlling. One could simply

use conventional materials (such as polymeric substrates) and create such “self-supporting chips” to achieve the same properties. What are the added advantages of these “complex water patterns”?

It is desirable to see how the proposed technique can overcome the current challenges in laser micromachining. For instance, one of the major problems in laser micromachining is that its resolution is diffraction limited, meaning that features are always limited to 2.5D (Ref.: <https://doi.org/10.1016/B978-1-78242-074-3.00022-2>). Can the proposed technique resolve this problem? Also, what is the minimum feature size that can be achieved using this technique? Does it improve the current minimum achievable size compared to other techniques?

Reviewer #3 (Remarks to the Author):

This is an interesting work and deserves publication in a highly reputed journal. Chips having a variety of patterns are fabricated using light from a liquid layer covered with hydrophobic nanoparticles. Proof of principle applications, such as liquid valves, droplet mixing, droplet arrays, chemical reactions, etc., are successfully demonstrated. Overall, this manuscript is an important contribution to microfluidics community and related fields. However, there are a few points I would like to clarify before recommending the same for publication.

1. I feel the title of this paper is misleading. The developed technology is not direct water machining. The machining is achieved by decorating the water layer with hydrophobic nanoparticles (liquid marble technology), and heating this layer using light. It is suggested to modify the title to reflect this aspect.
2. Achieved channel dimensions are about 1 mm or higher (figure 3h and other figures). Is it possible to achieve channels having lower dimensions?
3. Is it possible to reuse the channels after performing a liquid manipulation experiment? Also, is it possible to transfer the patterns without deformations?
4. Details of the laser and the optical setup will help the readers to repeat the experiment. Whether the same mechanism will work if the liquid is light absorbing?

Responses to Reviewer #1:

This paper presents a new method for direct machining and flexible patterning of water by using laser. The developed technique was successfully applied to some relevant microfluidic applications including liquid manipulation, chemical synthesis, biochemical analysis, liquid metal manipulation, conductive hydrogel synthesis, cell culture, and drug screening. The paper is well written and well organized. The results obtained should be of great interest to those working in related fields as well as in the wider field. Therefore, I think this paper can be published in Nature Communications if the authors can properly address my comments listed below.

We thank the reviewer for the positive comments on our manuscript.

Comment #1: *Laser cutting mechanism is unclear. Why can the NP film enhance the absorption. SiO₂ has absorption only in VUV (shorter than 190 nm) and IR (longer than 3 μm). What kind of laser was used? In this context, more details of laser cutting system (wavelength, pulse width or CW, focusing lens (NA), spot size, etc.) and irradiation conditions (power density or fluence, scanning speed, etc.) should be described in Methods.*

Response: We thank the reviewer for these comments. The laser used in our work is a CO₂ laser with an infrared light with wavelength of 10.6 μm. To test the absorption ability of the SiO₂ NPs used in our work, we characterized the infrared spectra of the SiO₂ NPs within the wavelength range of 2.5-20.0 μm. As shown in **Supplementary Fig. 1**, the SiO₂ NPs have absorption at the wavelength of 10.6 μm. To better explain the mechanism of laser cutting of nanoparticle-encased water pancakes (LCNEWP), we have added the description of the absorption capacity of SiO₂ NPs under infrared light with wavelength of 10.6 μm in our revised manuscript and supplementary information. In addition, we have also added the detailed experimental parameters of LCNEWP in the Methods section of our revised manuscript, as described below.

Supplementary Figure 1. Infrared spectra of the SiO₂ nanoparticles (NPs).

(Page S3, Supplementary information)

“Supplementary Note 1: Infrared spectra of the SiO₂ NPs

To test the absorption ability of the SiO₂ nanoparticles (NPs) used in our work for infrared light with a wavelength of 10.6 μm, we characterized the infrared spectra of the SiO₂ NPs collected from the nanoparticle-encased water pancake (NEWP) in the wavelength range of 2.5–20.0 μm (Supplementary Figure 1).”

(Page S10, Supplementary information)

“We next developed a theoretical model to reveal the mechanism of LCNEWP (Fig. 2a), involving the two main physical processes, *i.e.*, water vaporization and water replenishment. First, after contacting with laser with wavelength of 10.6 μm, the SiO₂ NPs on the surface of the water pancake absorb laser energy (Supplementary Fig. 1). Then, the temperature of SiO₂ NPs increases and thermal energy is transferred to the underlying water, causing the vaporization of water around the laser irradiation spot. Meanwhile, the surrounding water would flow towards the laser irradiated location to replace the vaporized water, bringing NPs to the exposed water surface. When the speed of water vaporization is greater than the speed of water replenishment, the NEWP can be cut through. As the hydrophobic NPs coat the exposed water surface again, the incision cannot be healed through water flow, leading to a clean cut through the NEWP.”

(Pages 5-6, main text)

“The CorelDRAW drawing software was used to design a line pattern of a chip

boundary. The self-supporting chips (SSCs) containing a small amount of liquid were obtained through directly engraving the line pattern onto a thin NEWP using a continuous-wave (CW) laser with wavelength of 10.6 μm created by a laser engraving machine (EMT-9060, EMIT laser, China). The laser cutting machine was equipped with a focusing lens with a focal length of 63.5 mm, which generated a laser spot with a diameter about 0.2 mm. The scanning speed of the laser was 10 mm s⁻¹ and the power density was 210 J mm⁻² during the cutting process of NEWP.”

(Pages 24-25, main text)

Comment #2: *Temperature of the NP film increased by laser irradiation and temperature distribution generated in water after laser irradiation should be simulated.*

Response: We thank the reviewer for this important suggestion. To better demonstrate the elevation of the temperature of the SiO₂ NPs film around laser irradiation point and the temperature distribution in water generated after laser irradiation, we have revised our COMSOL model and added the following analysis about the simulation results in our revised manuscript.

“Second, to simulate the LCNEWP process, we analyzed the heat transfer and the shape evolution of a NEWP during LCNEWP using finite element analysis (Fig. 2c, 2d, Supplementary Videos 1-2). After exposure to laser, the temperature of SiO₂ NPs around laser irradiation point gradually increases to over one thousand Celsius degrees owing to the absorption of laser energy (Fig. 2c). Then, the SiO₂ NPs transfer the heat to water, causing the temperature increase of water in the heated area. As the temperature of water reaches the boiling point (*i.e.*, 100 °C), the water would gradually vaporize and become thinner (Fig. 2d). Under the condition of constant laser heating, the time consumed to cut NEWP with a water thickness of 1.00 mm is about tenfold that of water with a thickness of 0.25 mm. A wider range of temperature and shape changes are generated within the water with a thickness of 1.0 mm due to the longer time of heat transfer, which confirms that thickness of water is a significant parameter that affects the LCNEWP process.”

(Page 8, main text)

Fig. 2. Theoretical analysis, numerical simulation and experimental optimization of parameters for laser cutting of nanoparticle-encased water pancakes. **a**, Schematic diagram of laser cutting of nanoparticle-encased water pancake (LCNEWP). **b**, Theoretical model of liquid flow during laser cutting. **c**, Simulation of temperature of the SiO₂ nanoparticles around the laser spot after exposure to the laser. **d**, Simulation of water shape evolution and temperature distribution during laser cutting, the initial thicknesses of the water in water pancake are 0.25 mm and 1.00 mm, respectively. **e**, Variation of nanoparticle-encased water pancake (NEWP) top view area with volume of water being pumped out (with an initial water volume of 25 mL). Data of distinct samples are presented as mean \pm s.d., $n = 3$. **f**, Effect of NEWP thickness and laser power on the outcome of LCNEWP. Brown symbols and background represent values at which the NEWP can be cut through by the laser. Blue symbols and background represent values at which the NEWP cannot be cut through by the laser. **g**, Effect of NEWP thickness and cutting speed on slit width. Data of distinct samples are presented as mean \pm s.d., $n = 3$. Insert image is the optical microscopic photograph of LCNEWP. (Page 10, main text)

“Finite element analysis on evolution of water shape and temperature distribution during laser cutting. The simulations of evolution of water shape and temperature distribution during laser cutting were conducted using COMSOL Multiphysics 6.0 (COMSOL, Sweden) (Fig. 2c,2d). In the finite element analysis procedure, the NEWP

is divided into two parts (*i.e.*, SiO₂ NPs film and water). First, the temperature distribution of SiO₂ NPs film is simulated using the solid heat transfer method, in which the derived temperature distribution of the SiO₂ NPs contacting the upper water surface is fitted into a Gauss function as shown in equation (12).

$$T(x) = T_0 + A * \exp\left(-\frac{2x^2}{\omega^2}\right) \quad (12)$$

where T and x refer to the temperature and position of the SiO₂ NPs, respectively. The parameters T_0 , A and ω in equation (12) vary with time t , and the fitted results are shown in equation (13).

$$\begin{cases} T_0(t) = -41 + 61 * \exp\left(-\frac{25t}{18}\right) \\ A(t) = 1938 - 1938 * \exp\left(-\frac{5t}{2}\right) \\ \omega(t) = 2.95 - 0.95 * \exp(-2t) \end{cases} \quad (13)$$

Second, the temperature of the upper surface of the water is set equal to the temperature of the underlying SiO₂ NPs, since the upper surface of the water is directly contacted with the underlying SiO₂ NPs. Then, the temperature distribution and the shape evolution of water with thicknesses of 0.25 mm and 1.00 mm were simulated by establishing a fluid heat transfer module and using a sensible enthalpy method, respectively.”

(Page 25, main text)

Comment #3: *Details on formation of liquid pancake (Fig. 1 a1-2) should be also described in Methods.*

Response: We thank the reviewer for this comment. We have added the detailed procedure of making liquid pancakes in the Methods section of our revised manuscript as follows.

“Preparation of NEWP. The preparation of the **NEWP** consisted of the following procedures: (1) adding 10.00 mL of SiO₂ NPs sol to a 15 cm×15 cm Petri dish; (2) shaking the Petri dish to coat the inner surface of the Petri dish with SiO₂ NPs sol; (3) recovering the remaining SiO₂ NPs sol; (4) standing at room temperature for 15 min; (5) adding 45.00 mL of water to the Petri dish, and (6) sucking 42.75 mL of water from the Petri dish.”

(Page 24, main text)

Comment #4: *Scale bars should be included in Fig. 1 b.*

Response: We thank the reviewer for this suggestion. We have added the scale bars in our revised **Fig. 1b** as below.

Fig. 1. Machining of nanoparticle-encased water pancakes through laser cutting. **a**, Schematic illustration of machining water through laser cutting of water pancake encased by silica nanoparticles (NPs). **b**, Drawings, Chinese characters and letters formed by injecting various liquids (ink, milk, and pigment) into water patterned by laser cutting. The scale bars represent 1 cm.

(Page 5, main text)

Comment #5: *In Fig. 2 f, why is the slit width approximately equal to the water thickness regardless of scan speed? Is the slit width larger than the spot size? Can smaller spot size create narrower slit? The inset photo shows the irregularities in the slit width. What is the standard deviation of the width? Will such irregularities affect*

the applications?

Response: We thank the reviewer for raising these questions. The figure number of **Fig. 2f** was replaced with **Fig. 2g** in the revised manuscript. First, as shown in **Fig. 2g**, the slit width created by laser cutting of nanoparticle-encased water pancakes (LCNEWP) is approximately 10 times of the water pancake thickness. The reason why the slit width becomes smaller as the water pancake thickness decreases is that the thinner water pancake can be cut in a shorter time period, and the heat generated by laser has a smaller lateral transfer in water, causing a smaller area of water to be vaporized. Thus, a smaller slit can be obtained with LCNEWP when the thickness of the water pancake is thinner. Second, the slit width (350 μm) created by LCNEWP is larger than the spot size (200 μm). During the process of LCNEWP, the excessive laser energy would directly heat the Petri dish under water pancake after the water pancake is cut. While the effect of laser on the water pancake is relatively small due to the slit width is larger than the spot size. Thus, under the constant laser power, the scanning speed of laser would not have significant effect on the slit width created by LCNEWP. Third, the smaller spot size allows for narrower slits. To check whether a smaller spot can achieve a smaller slit of LCNEWP, we adjusted the focusing lens of the laser cutting system with a smaller focal length (50.8 mm), through which a smaller laser spot and slits with a width of 200 μm were obtained. Fourth, we achieved a more regular slit relative to the one in the inset photo and updated it in **Fig. 2g**. The standard deviation of the updated slit width is about 20 μm . Last, the irregularity of the slit may not have a significant impact on the application of self-supporting chips, since the standard deviation of the slit width is less than 6% of the width of the smallest channel that can be processed (350 μm). To clarify these, we have added the following sentences in our revised manuscript.

“Fourth, to obtain the optimal laser cutting results, slit width was used as an index for evaluation of laser cutting and optimization of parameters that affect machining accuracy including NEWP thickness, cutting speed and spot size. As shown in **Fig. 2g**, the less NEWP thickness corresponds to narrower slits produced during LCNEWP and thereby higher machining accuracy. **The reason why the slit width becomes smaller as the NEWP thickness decreases is that the thinner NEWP can be cut in a shorter period of time, and the heat generated by laser has a smaller lateral transfer in water, causing a smaller area of water to be vaporized. Thus, a smaller slit can be obtained with**

LCNEWP when the NEWP thickness is thinner. As such, NEWP with an optimized thickness of 0.1 mm were used for subsequent experiments. Furthermore, during the process of LCNEWP, the excessive laser energy would heat the Petri dish under NEWP after the NEWP is cut. While the effect of laser on the NEWP is relatively small due to the slit width is larger than the spot size. Therefore, as shown in Fig. 2g, under the condition that LCNEWP can be realized, the cutting speed did not significantly impact the machining accuracy within a certain range. For the effect of the spot size on the slit width, a laser machining straight line with slit width of 350 μm can be obtained using a laser with a spot diameter of 200 μm (Fig. 2g), while the smaller spot size allows for a slit width of 200 μm (Supplementary Fig. 5). That is, the NEWP thickness and the spot size are the key parameters that affect the resolution of LCNEWP.”

(Pages 8-9, main text)

Fig. 2. Theoretical analysis, numerical simulation and experimental optimization of parameters for laser cutting of nanoparticle-encased water pancakes. **a**, Schematic diagram of laser cutting of nanoparticle-encased water pancake (LCNEWP). **b**, Theoretical model of liquid flow during laser cutting. **c**, Simulation of temperature of the SiO₂ nanoparticles around the laser spot after exposure to the laser. **d**, Simulation of water shape evolution and temperature distribution during laser cutting, the initial

thicknesses of the water in water pancake are 0.25 mm and 1.00 mm, respectively. **e**, Variation of **nanoparticle-encased water pancake (NEWP)** top view area with volume of water being pumped out (with an initial water volume of 25 mL). Data of distinct samples are presented as mean \pm s.d., $n = 3$. **f**, Effect of NEWP thickness and laser power on the outcome of **LCNEWP**. Brown symbols and background represent values at which the NEWP can be cut through by the laser. Blue symbols and background represent values at which the NEWP cannot be cut through by the laser. **g**, Effect of NEWP thickness and cutting speed on slit width. Data of distinct samples are presented as mean \pm s.d., $n = 3$. Insert image is the optical microscopic photograph of **LCNEWP**.
(Page 10, main text)

Supplementary Figure 5. Optical microscope photograph of a slit created by laser cutting of nanoparticle-encased water pancakes (LCNEWP) and a microscope eyepiece scale. The width of the slit enlarged by 50 folds is about 1 cm. The laser cutting system was equipped with a focusing lens with a focal length of 50.8 mm.

(Page S5, Supplementary information)

“Supplementary Note 5: Photograph of a slit created by laser cutting of nanoparticle-encased water pancakes (LCNEWP)

To investigate whether a smaller spot can achieve a smaller slit, we adjusted the focal length of the laser cutting system through using a focusing lens with a smaller focal length (50.8 mm) to obtain a smaller laser spot. We can see that the slits with a width of 200 μm can be achieved through using the focusing lens with a smaller focal length (**Supplementary Figure 5**).”

(Page S11, Supplementary information)

Comment #6: *The authors mentioned some advantages of SSC over the conventional solid microfluidic chips. What are the disadvantages of SSC? Fabrication resolution (accuracy of 350 μm) is much worse than the fabrication technique of conventional*

solid microfluidic chips. 3D microfluidic structures cannot be fabricated. Please comment on these.

Response: We thank the reviewer for raising these questions. The laser machine used in our work can achieve the smallest fabrication resolution of the laser cutting of nanoparticle-encased water pancakes (LCNEWP) as 200 μm . And since the SSCs are maintained in shape by the self-supported liquid, it is difficult for SSCs to keep shape in the vertical direction of the 3D structure due to the effect of gravity. We thus agree with the reviewer that the accuracy of processing and the construction capability of 3D structure of the SSCs proposed in our work are still not comparable to those of the conventional microfluidic chips. In our future study, we can use the laser with a smaller spot size and reduce the mobility of water to further improve the accuracy of SSCs. To clarify these, we have added the following sentences in our revised manuscript.

“The laser machine used in our work can achieve the smallest fabrication resolution of the LCNEWP as 200 μm . And since the SSCs are maintained in shape by the self-supported liquid, it is difficult for the SSCs to keep shape in the vertical direction of the 3D structure due to the effect of gravity. The accuracy of processing and the construction capability of 3D structure of the SSCs proposed in our work are not comparable to those of the conventional microfluidic chips. But we can use the laser with a smaller spot size and reduce the mobility of water to further improve the accuracy of SSCs in our future study.”

(Page 23, main text)

Comment #7: *The authors said “in terms of industry, the precise machining of water through laser cutting offers far simplified and lower material consumption pathway for patterning water compared with the traditional method of processing pre-patterned containers, which can reduce time and material costs in the field of microfluidic, especially”. However, solid microfluidic chips can be currently mass-produced by the embossing technique with cheap costs. The authors should comment on this.*

Response: We thank the reviewer for raising this question. We agree with the reviewer that the conventional microfluidic chips can be machined by mass production at low cost. While, compared with the traditional microfluidic chip processing technology (e.g., lithography and embossing), the technology of direct machining SSCs by laser

cutting of nanoparticle-encased water pancakes (LCNEWP) proposed in our work allows for flexible processing of microfluidic chips with different structures due to no need of the procedure of making molds. Thus, the technology of direct machining SSCs by laser cutting of nanoparticle-encased water pancakes (LCNEWP) can satisfy the demands of personalized microfluidic chips. In addition, the LCNEWP proposed in this study has the application potential of industrial processing of SSCs due to the advantages of digital control and high speed of laser cutting technology. To clarify these, we have revised our statement in our revised manuscript as follow.

“Moreover, in terms of industry, the precise machining of water **in NEWP** through laser cutting, **which has the advantages of flexible design and fast machining, has the application potential in the industrial processing of personalized microfluidic chips.**”

(Page 23, main text)

Responses to Reviewer #2:

In this study, the authors have proposed a method for laser micromachining of water coated with hydrophobic nanoparticles. They have showcased the applications of this technique by fabricating various self-supporting chips.

We thank the reviewer for the comments on our manuscript.

Comment #1: *Firstly, the title of this manuscript is misleading. While the title looks very promising for Nature Communication journal, the main proposed technique does not offer direct laser micromachining of water. In fact, the authors have laser machined the “liquid marble” film. They refer to this liquid marble as “non-wetted liquid pancake”. Also, in page 4 of the manuscript, it is mentioned that more than 95% of the enclosed water had been withdrawn from the “liquid pancake”. As such, the method is no longer “directly” machining the water. Since the SiO₂ nanoparticles are transparent, it creates the illusion that the laser is cutting the water, which is scientifically not correct.*

Response: We thank the reviewer for raising this question. Following this comment, we have revised the title and the corresponding statements in our revised manuscript as below.

“**Machining water through laser cutting of nanoparticle-encased water pancakes**”

(Page 1, main text)

“Here we report a strategy through laser cutting to realize the machining of **nanoparticle-encased water pancakes** when water surface is coated with SiO₂ nanoparticles and depth of water is at sub-millimeter level. The process of **laser cutting of nanoparticle-encased water pancakes** and the parameters that affect the cutting accuracy are supported by theoretical analysis, numerical simulation and experimental studies.”

(Page 2, main text)

“Herein, we developed a strategy for machining water through laser cutting of **nanoparticle-encased water pancakes (LCNEWP)**, in which water was coated with SiO₂ nanoparticles and manipulated to form a water pancake with sub-millimeter thickness. With this strategy, water **pancake** can be cut and a submillimeter-scale cutting slit can

be generated. The process of **LCNEWP** and the parameters that affect the cutting accuracy were well investigated through theoretical analysis, numerical simulation and experimental study.”

(Pages 3-4, main text)

“Machining of **nanoparticle-encased water pancakes** by laser cutting”

(Page 4, main text)

“Greater than 95% of the water was then withdrawn from the **nanoparticle-encased water pancake (NEWP)**.”

(Page 4, main text)

“Then we demonstrated this strategy to fabricate various patterns (including drawings, Chinese characters and letters) through **LCNEWP** and injection of colored liquids (ink, milk and pigment) (**Fig. 1b**).”

(Page 5, main text)

“Fig. 1. Machining of nanoparticle-encased water pancakes through laser cutting. a, Schematic illustration of machining water through laser cutting **of water pancake encased by silica nanoparticles (NPs).** **b,** Drawings, Chinese characters and letters formed by injecting various liquids (ink, milk, and pigment) into water patterned by laser cutting. **The scale bars represent 1 cm.”**

(Page 5, main text)

“Theoretical analysis, numerical simulation and optimized experimental parameters for laser cutting of **nanoparticle-encased water pancakes**”

(Page 5, main text)

“Various self-supporting chips fabricated through **laser cutting of nanoparticle-encased water pancakes**”

(Page 10, main text)

“In view of the digital control, simple operation and fast characteristics of the laser cutting technology, the **LCNEWP** methodology can be applied as a high-throughput strategy for machining water **in NEWP**. To demonstrate the aforementioned advantages, a computer-controlled laser cutting procedure for machining water **in NEWP** is shown

in **Fig. 3a** and **Supplementary Video 3**, in which a closed-loop pattern that successfully isolated the water inside the incision was generated in 12 seconds. To demonstrate the capability of laser cutting as a general method for machining water in **NEWP**, a series of commonly used water patterns were processed.”

(Pages 10-11, main text)

“**Fig. 3. Various self-supporting chips (SSCs) fabricated through laser cutting of nanoparticle-encased water pancakes.** **a**, Dynamic process of the SSC fabrication through laser cutting of nanoparticle-encased water pancakes. **b**, Cross-type SSC chip. **c**, A partial enlarged view of a straight channel and a microscope eyepiece scale. The diameter of the straight channel enlarged by 50 folds is about 1.75 cm. **d**, SSC with radial array fluid channels. **e**, Droplet array SSC. **f**, SSC with a single curved fluid channel. **g**, SSC with multiple fluid channels. **h**, SSC with array of curved fluid channels. **i**, Spiral SSC. The scale bars represent 1 cm.”

(Pages 12-13, main text)

“The strategy that machining water in **NEWP** through laser cutting with using water coated with SiO₂ NPs and with submillimeter thickness is proposed to minimize the effect of properties of water and laser as the machining tool, with which the patterning and self-supporting shapes of water in **NEWP** were achieved.”

(Page 22, main text)

“Furthermore, the theoretical analysis, numerical simulation and experimental study for the process of **LCNEWP** and the parameters that affect the cutting accuracy can provide inspiration for researchers focusing on the processing of other liquid materials.”

(Pages 22-23, main text)

“We thus expect the cooperation with professionals working on physics, biology, medicine, chemistry, materials, and other interdisciplinary fields to promote further development of the strategy of machining water in **NEWP** with laser cutting.”

(Page 23, main text)

“Operating procedures for fabrication of SSCs through **LCNEWP**.”

(Page 24, main text)

“First, the SSCs used in the fluid manipulation demos (**Fig. 4a-j**) were all manufactured

through LCNEWP.”

(Page 25, main text)

Comment #2: *Secondly, while the proposed technique looks interesting, the advantages of this strategy are not clear. There are no added advantages to creating this substrate for laser cutting. For example, liquid manipulation in “self-supporting chips” still needs conventional valves and external pumps. So, why one should make the fabrication overcomplicated by coating the water with hydrophobic SiO₂ nanoparticles. There are many health concerns and safety hazards in handling these nanoparticles, as well. They listed the following items as the main advantages of these platforms: openness, transparency, breathability, liquid morphology and flow controlling. One could simply use conventional materials (such as polymeric substrates) and create such “self-supporting chips” to achieve the same properties. What are the added advantages of these “complex water patterns”?*

Response: We thank the reviewer for these comments. Besides similar capabilities to the conventional microfluidic chips made of polymeric materials in terms of fluid manipulation, the self-supporting chips (SSCs) developed in our work also possess some unique properties. For example, the bubble issue that normally occurs in the closed microfluidic chips can be easily overcome using the open solid wall-free microfluidic chips (*Nature Communications*, 2017, 8(1): 816). The open microfluidic chips without solid walls can avoid the high shearing force, which might damage cells, proteins and antibodies, of the closed microfluidic chips during fluid manipulation (*Nature*, 2020, 581(7806): 58-62). For valves, benefiting from the cuttability and connectability of SSCs, realization of the valve function of SSCs does not require a complex valve structure as that of a closed microfluidic chip. In addition, the liquid flow can be driven by the pressure provided by gravity, thus the pump to drive the flow of liquid inside the solid channel of the closed microfluidic chip is not essential for the SSCs. To clarify these, we have added the description about the self-driven flow process of liquids dropped through a syringe in Supplementary Information.

“However, the pump is not essential for SSCs since the flow of liquid can be driven by the pressure provided by the gravity of liquid (**Supplementary Fig. 10, Supplementary Video 6**). The valve function was realized through exploiting the SSC’s properties of openness, cuttability, and connectivity, **which is not a complex**

valve structure as that of a closed microfluidic chip.”

(Page 13, main text)

Supplementary Figure 10. Photographs of the flow process of red liquid driven by the pressure provided by the own gravity of liquid in a self-supporting chip. The scale bar represents 1 cm.

(Page S7, Supplementary information)

“Supplementary Note 10: Self-driving liquid flow within self-supporting chip

For SSCs, the flow of the liquid can be driven by the pressure provided by the liquid's gravity. The self-driven flow process of the red liquid dropped through a syringe, in which the red liquid was dropped vertically into the self-supporting chip as demonstrated in **Supplementary Figure 10.**”

(Page S12, Supplementary information)

Second, fabrication of anisotropic functional patterned surfaces is the main method to create open microfluidic chips due to its advantages of openness, transparency, breathability, liquid morphology and flow controlling (*Nature Communications*, 2022, 13(1): 3078; *Angewandte Chemie International Edition*, 2020, 59(26): 10535). However, surface patterning of solid substrates is a complicated procedure and often needs expensive moulds/masks and photoresists. Manufacturing self-supporting chips (SSCs) by laser cutting of nanoparticle-encased water pancakes (LCNEWP) can avoid the complex procedure of processing the substrates. Moreover, SSCs have the capability to isolate the liquid inside the SSCs from the particulate contaminants in surrounding environment since the liquid is encapsulated by the nanoparticle film on the surface of SSCs. Whereas, the liquid in an open microfluidic chip fabricated by patterning the surface of a solid substrate is exposed to surrounding environment and is

susceptible to contamination. To emphasize these, we have added the following sentences in our revised manuscript.

“In addition, the bubble issue that normally occurs in the closed microfluidic chips can be easily overcome using the open solid wall-free microfluidic chips¹⁴. Besides, the open microfluidic chips without solid walls can avoid the high shearing force of the closed microfluidic chips during fluid manipulation, which might damage cells, proteins and antibodies¹⁹. SSCs also have the capability to isolate the liquid inside the SSCs from the particulate contaminants in the surrounding environment since the liquid is encapsulated by the nanoparticle film on the surface of SSCs.”

(Page 12, main text)

“14. Walsh, E. J., Feuerborn, A., Wheeler, J. H., Tan, A. N., Durham, W. M., Foster, K. R., & Cook, P. R. Microfluidics with fluid walls. *Nat. Commun.* **8**, 816 (2017).

19. Dunne, P. et al. Liquid flow and control without solid walls. *Nature* **581**, 58–62 (2020).”

(Pages 31-32, main text)

Comment #3: *It is desirable to see how the proposed technique can overcome the current challenges in laser micromachining. For instance, one of the major problems in laser micromachining is that its resolution is diffraction limited, meaning that features are always limited to 2.5D (Ref.: <https://doi.org/10.1016/B978-1-78242-074-3.00022-2>). Can the proposed technique resolve this problem? Also, what is the minimum feature size that can be achieved using this technique? Does it improve the current minimum achievable size compared to other techniques?*

Response: We thank the reviewer for raising these questions. We agree with the reviewer that diffraction as an important factor limits the resolution of laser micromachining technique and causes that the fabricated features are limited to 2.5D. Our work aims to exploring and addressing the challenge of how laser cutting technology can be used to machine liquid. Firstly, the laser processing-based technologies (e.g., laser cutting, engraving, lithography and printing), with the advantages of high precision, fast speed, and operational simplicity, have been widely employed in processing of solid materials. But micromachining of liquids with fluidity and light transmission by laser cutting technology is still challenging. Our strategy of

the laser-based precisely machining of nanoparticle-encased water pancake (NEWP) can overcome the challenge that liquid is difficult to be processed using laser cutting. Moreover, the theoretical analysis, numerical simulation and experimental study of the LCNEWP process and the parameters affecting the cutting accuracy can provide references for the laser-based processing of other liquids.

Secondly, as for the minimum feature size that can be achieved using our method, we can fabricate the SSCs with the minimum size of 350 μm . The resolution of the SSCs processed by laser cutting is similar to that of the open microfluidic chips constructed using liquid walls (*Nature Communications*, 2017, 8(1): 816), and is better than that of mechanical shaping of liquid plasticine/pancake in previous reports (*Advanced Materials Interfaces*, 2020, 7(24): 2001573; *Advanced Materials Interfaces*, 2018, 5(2): 1701139). Even the resolution of the SSCs made by laser cutting is still lower than that of the conventional solid-state microfluidic chips, it may be further enhanced through using laser with smaller spot and reducing the fluidity of water. To clarify these, we have added the following sentences in our revised manuscript.

“Here we report a strategy through laser cutting to realize the machining of nanoparticle-encased water pancakes when water surface is coated with SiO₂ nanoparticles and depth of water is at sub-millimeter level.”

(Page 2, main text)

“This work provides a strategy for addressing a challenge of laser cutting to precisely machine water, which would arouse interest in widespread fields involving fluid patterning and flow control in biological, chemical, materials and biomedical researches.”

(Page 2, main text)

“Herein, we developed a strategy for machining water through laser cutting of nanoparticle-encased water pancakes (LCNEWP), in which water was coated with SiO₂ nanoparticles and manipulated to form a water pancake with sub-millimeter thickness.”

(Page 3, main text)

“Our work provides a strategy for overcoming the challenge of laser cutting to precisely machine water, and a path to achieve water patterning in open environment in an easy

and rapid manner, which holds a wide range of application potentials in chemistry, biology, materials science, and biomedicine.”

(Page 4, main text)

“This manuscript provides a strategy to address a challenge of laser cutting to precisely machine water, which is caused by the inherent properties of water such as fluidity and disorder.”

(Page 22, main text)

“From the perspective of physical science, the effective strategy for laser-based precise machining of water in NEWP is a contribution for overcoming the challenge that water is difficult to cut using laser.”

(Page 22, main text)

“And the resolution of the formed straight channel reaches 350 μm (Fig. 3c). The resolution of the SSCs processed by laser cutting is similar to that of the open microfluidic chips constructed using liquid walls¹⁴, and is better than that of mechanical shaping of liquid plasticine/pancake in previous reports^{15,16}. Even the resolution of the SSCs made by laser cutting is still lower than that of the conventional solid-state microfluidic chips, it can be further enhanced through using laser with the smaller spot and reducing the fluidity of water.”

(Page 11, main text)

“14. Walsh, E. J., Feuerborn, A., Wheeler, J. H., Tan, A. N., Durham, W. M., Foster, K. R., & Cook, P. R. Microfluidics with fluid walls. *Nat. Commun.* **8**, 816 (2017).

15. Li, X. et al. Liquid shaping based on liquid pancakes. *Adv. Mater. Interfaces* **5**, 1701139 (2018).

16. Fujiwara, J., Geyer, F., Butt, H. J., Hirai, T., Nakamura, Y., & Fujii, S. Shape-designable polyhedral liquid marbles/plasticines stabilized with polymer plates. *Adv. Mater. Interfaces* **7**, 2001573 (2020).”

(Page 31, main text)

Responses to Reviewer #3:

This is an interesting work and deserves publication in a highly reputed journal. Chips having a variety of patterns are fabricated using light from a liquid layer covered with hydrophobic nanoparticles. Proof of principle applications, such as liquid valves, droplet mixing, droplet arrays, chemical reactions, etc., are successfully demonstrated. Overall, this manuscript is an important contribution to microfluidics community and related fields. However, there are a few points I would like to clarify before recommending the same for publication.

We thank the reviewer for the positive comments on our manuscript.

Comment #1: *I feel the title of this paper is misleading. The developed technology is not direct water machining. The machining is achieved by decorating the water layer with hydrophobic nanoparticles (liquid marble technology), and heating this layer using light. It is suggested to modify the title to reflect this aspect.*

Response: We thank the reviewer for this suggestion. Following this comment, we have revised the title of our revised manuscript as below.

“Machining water through laser cutting of nanoparticle-encased water pancakes”

(Page 1, main text)

Comment #2: *Achieved channel dimensions are about 1 mm or higher (figure 3h and other figures). Is it possible to achieve channels having lower dimensions?*

Response: We thank the reviewer for raising this question. We can obtain the water channel with dimension smaller than 1 mm through laser cutting in our work. For example, a straight channel with dimension of 350 μm can be achieved. To clarify this, we have added the photos of the fabricated channels (**Fig. 3c**) and the corresponding descriptions in our revised manuscript as below.

Fig. 3. Various self-supporting chips (SSCs) fabricated through laser cutting of nanoparticle-encased water pancakes. **a**, Dynamic process of the SSC fabrication through laser cutting of nanoparticle-encased water pancake. **b**, Cross-type SSC chip. **c**, A partial enlarged view of a straight channel and a microscope eyepiece scale. The diameter of the straight channel enlarged by 50 folds is about 1.75 cm. **d**, SSC with radial array fluid channels. **e**, Droplet array SSC. **f**, SSC with a single curved fluid channel. **g**, SSC with multiple fluid channels. **h**, SSC with array of curved fluid channels. **i**, Spiral SSC. The scale bars represent 1 cm.

(Pages 12-13, main text)

“And the resolution of the formed straight channel reaches 350 μm (Fig. 3c). The resolution of the SSCs processed by laser cutting is similar to that of the open microfluidic chips constructed using liquid walls¹⁴, and is better than that of mechanical shaping of liquid plasticine/pancake in previous reports^{15,16}. Even the resolution of the SSCs made by laser cutting is still lower than that of the conventional solid-state microfluidic chips, it may be further enhanced through using laser with smaller spot

and reducing the fluidity of water.”

(Page 11, main text)

Comment #3: *Is it possible to reuse the channels after performing a liquid manipulation experiment? Also, is it possible to transfer the patterns without deformations?*

Response: We thank the reviewer for raising these questions. The self-supporting chips (SSCs) fabricated in our work can be reused after performing liquid manipulation experiments. For example, we can replace the solution in the channel with water by multiple rinses. We have tried the experiments of placing the SSC on a shaker with a speed of 100 rpm. The results show that the SSC was stable when being transferred along with the substrate without obvious deformation, *i.e.*, the pattern during transfer was stable. We also agree with the reviewer that it is difficult to peel the SSCs off the substrate without deformation. To clarify these, we have added the new experimental results in our revised supplementary information and added the corresponding discussions in our revised manuscript.

“Furthermore, the SSCs can be reused after performing liquid manipulation experiments (Supplementary Fig. 11).”

(Page 14, main text)

“After forming a pattern consisted of NPs film and encapsulated water, various self-supporting chips (SSCs) with stable shapes in open environments without solid walls were created, which present no obvious deformation during transfer with the substrate (Supplementary Fig. 6, Supplementary Video 4).”

(Page 11, main text)

Supplementary Figure 11. Photographs of the replacement process of red liquid in a self-supporting chip with water by rinses for multiple times. The scale bar represents 1 cm.

(Page S8, Supplementary information)

“Supplementary Note 11: Reuse of self-supporting chips

The SSCs can be reused after performing liquid manipulation experiments. The replacement of the red liquid in the SSC with purified water by multiple rinses, in which rinsing of the self-supporting chip was performed by sequentially injecting pure water into the chip and sucking liquid from the chip (**Supplementary Figure 11**).”

(Pages S12-S13 Supplementary information)

Supplementary Figure 6. Photos of the self-supporting chips with and without shaking on a shaper with a shaking frequency of 100 rpm. The scale bar represents 1 cm.

(Page S5, Supplementary information)

“Supplementary Note 6: Stability of self-supporting chips during transfer

To explore the stability of the self-supporting chip (SSC) during transfer, the SSC was placed on a shaker with a speed of 100 rpm. The photos of the SSCs with and without shaking are shown in **Supplementary Figure 6.**”

(Page S11, Supplementary information)

Comment #4: *Details of the laser and the optical setup will help the readers to repeat the experiment. Whether the same mechanism will work if the liquid is light absorbing?*

Response: We thank the reviewer for the suggestion and question. Following this comment, we have added the experimental parameters of the laser cutting of nanoparticle-encased water pancakes (LCNEWP) process in the Methods section of our revised manuscript. To explore whether the same mechanism works for the liquid is light absorbing, we added the water-soluble SiO₂ nanoparticles, which can absorb infrared light with wavelength of 10.6 μm, to water to make the water pancake. Then the pentagram and circle patterns were machined using the LCNEWP system, showing that the laser cutting method also works for the light-absorbing liquids. Moreover, the laser power required to cut light-absorbing liquids is lower than that of pure water. To clarify these, we have added the experimental results and the corresponding discussions in our revised manuscript and supplementary information.

“The CorelDRAW drawing software was used to design a line pattern of a chip boundary. The self-supporting chips (SSCs) containing a small amount of liquid were obtained through directly engraving the line pattern onto a thin NEWP using a continuous-wave (CW) laser with wavelength of 10.6 μm created by a laser engraving machine (EMT-9060, EMIT laser, China). The laser cutting machine was equipped with a focusing lens with a focal length of 63.5 mm, which generated a laser spot with a diameter about 0.2 mm. The scanning speed of the laser was 10 mm s^{-1} and the power density was 210 J mm^{-2} during the cutting process of NEWP.”

(Pages 24-25, main text)

“In addition, the LCNEWP strategy is also suitable for the light-absorbing liquids, which require less laser energy during laser cutting than those of pure water (Supplementary Fig. 4).”

(Page 8, main text)

Supplementary Figure 4. Machining of light-absorbing liquid in nanoparticle-encased water pancake (NEWP) through laser cutting. (a) Comparison of laser power required for laser cutting between pure water and water containing SiO₂ NPs. (b) Pentagram pattern processed by laser cutting. (c) Circular pattern processed by laser cutting. The scale bars represent 5 mm.

(Page S4, Supplementary information)

“**Supplementary Note 4: Machining of light-absorbing liquid in nanoparticle-**

encased water pancake (NEWP) through laser cutting

To explore whether the same mechanism works when the liquid is light absorbing, the water used to make the NEWP was added with 0.2 g L^{-1} of water-soluble SiO_2 nanoparticles which absorb infrared light with a wavelength of $10.6 \text{ }\mu\text{m}$. The comparison of laser power required for cutting between water and water containing SiO_2 NPs is shown in **Supplementary Figure 4a**, from which we can see that the laser cutting method proposed in this study also works for the light-absorbing liquids (**Supplementary Figure 4b,c**).”

(Pages S10-11, Supplementary information)

Reviewer comments, second round

Reviewer #1 (Remarks to the Author):

The authors well addressed all of my comments, so this paper can be accepted for publication in Nature Communications.

Reviewer #2 (Remarks to the Author):

The authors have carefully addressed all of the comments and revised their manuscript accordingly. I believe the revised manuscript can be accepted now.

Reviewer #3 (Remarks to the Author):

The authors have addressed the queries and modified the draft. I believe the present version of the manuscript is publishable in Nature Communications.

Response to Reviewer #1:

Comment: *The authors well addressed all of my comments, so this paper can be accepted for publication in Nature Communications.*

Response: We thank the positive comments and encouragement from the reviewer on our revised manuscript.

Response to Reviewer #2:

Comment: *The authors have carefully addressed all of the comments and revised their manuscript accordingly. I believe the revised manuscript can be accepted now.*

Response: We thank the positive comments and encouragement from the reviewer on our revised manuscript.

Response to Reviewer #3:

Comment: *The authors have addressed the queries and modified the draft. I believe the present version of the manuscript is publishable in Nature Communications.*

Response: We thank the positive comments and encouragement from the reviewer on our revised manuscript.